# Pretraining LLM with Latent Thoughts in Continuous Space

## Abstract

The remarkable success of Chain-of-Thought (CoT), which enhances performance by scaling generation steps at test-time, inspires us to ask: can we leverage a similar scaling of computational steps during pretraining to improve the generation of each individual token? To address this, we propose a novel pre-training methodology: *Pretraining Language Models with Latent Thoughts*. Our approach pretrains a language model (LM) to first generate an intermediate latent thought—the last hidden state of the current position—which is then used as input to predict the actual subsequent token. This additional computational step enables the LM to refine its prediction within unconstrained continuous space. Our experiments demonstrate that, at an identical inference cost, a LM that generates one additional latent thought per token outperforms a standard model with *double* the parameters. For instance, ours-1.4B (Pythia Arch), pretrained on 300B tokens from the Pile, significantly surpasses the vanilla Pythia-2.8B trained on the same data on both language modeling and a range of general downstream tasks. Furthermore, increasing the number of latent thoughts generated before each actual token—forming a chain analogous to CoT—consistently improves the model's performance.

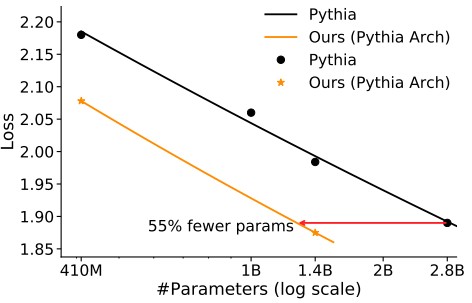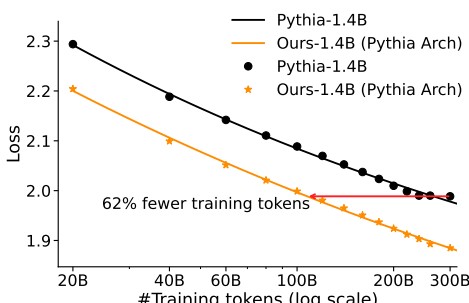

Figure 1: Scaling curves comparing our method (Pythia Arch) with the official Pythia suite on the 300B Pile. Our 1.26B model matches the loss of Pythia-2.8B with 55% fewer parameters (left), while our 1.4B model reaches the baseline's final performance with 62% less training data (right).

## 1 Introduction

The conventional wisdom in improving language models—scaling up parameters and data—is facing diminishing returns due to data scarcity (Villalobos et al., 2022; Muennighoff et al., 2023), saturating scaling laws (Hoffmann et al., 2022a; Hackenburg et al., 2025), and prohibitive training overheads (Pati et al., 2023; Narayanan et al., 2021; Li et al., 2024).

This has shifted focus towards enhancing model capabilities via test-time scaling (Snell et al., 2024), particularly through methods based on Chain-of-Thought (CoT) (Jaech et al., 2024; DeepSeek-AI et al., 2025). CoT achieves remarkable success by generating long reasoning chains for each question, effectively scaling the generation steps and increasing computation per query. While effective, CoT relies on specialized datasets and complex training schemes (Allen-Zhu & Li, 2023; Li et al.,

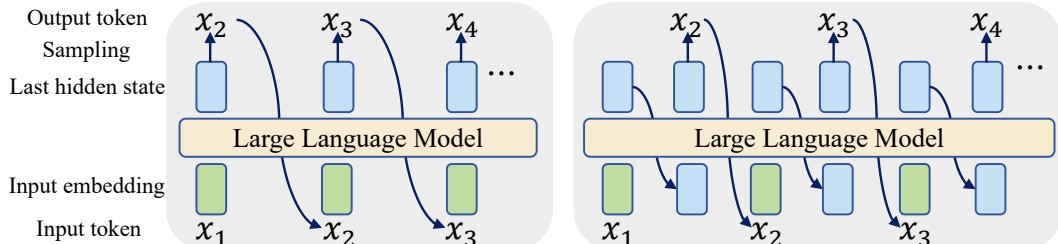

Figure 2: A comparison between the standard language model and ours. In the standard language model, each token is generated after a single forward pass. In contrast, ours does not immediately sample the output token after one forward pass; instead, it uses the computed last hidden state as the next input embedding for generating the subsequent output token. This allows the language model to think in an unconstrained latent space before producing each token.

2025; Pang et al., 2025), is confined to a discrete token space, and is ultimately capped by the base model's capabilities (Yue et al., 2025).

An alternative direction is to scale computation during pretraining. One approach, often termed "*vertical scaling*", deepens the network by reusing parameters (Zeng et al., 2025; Giannou et al., 2023; Geiping et al., 2025; Chen et al., 2025b). However, this can lead to training instabilities (Geiping et al., 2025) and often fails to outperform a standard dense model with a comparable inference budget, limiting its practical utility.

Inspired by the success of CoT in scaling generation steps, we propose a novel "*horizontal scaling*" approach: Pretraining Language Models with Latent Thoughts. Instead of deepening the model, our method teaches the LM to scale the generation process for each token. It first generates an intermediate latent thought—the last hidden state of the current position—which is then used as input to predict the actual subsequent token. This allows the model to refine its predictions in an unconstrained continuous space. To maintain training efficiency, we employ the Jacobi iteration (Saad, 2003; Barrett et al., 1994) to parallelize this inherently sequential process.

Our experiments show that, at an identical inference cost, a model trained with one latent thought per token surpasses a standard model with double the parameters. For instance, our 1.4B models, built on Pythia and LLaMA architectures, significantly outperform their vanilla 2.8B counterparts trained on the same data. Our method also proves superior to previous vertical scaling techniques, even when their inference cost is twice as high. Furthermore, increasing the number of latent thoughts to form a chain of latent thoughts—analogous to CoT—before generating each real token consistently improves model performance, further underscoring the potential of our approach.

## 2 RELATED WORK

We begin by discussing the two most related works: Coconut (Hao et al., 2024) and PonderLM (Zeng et al., 2025). Coconut *finetunes* a language model on *CoT data*, employing a "Chain of Continuous Thought"—represented by the final hidden states—to simulate explicit reasoning steps. This continuous chain is typically applied only after a question is posed. In contrast, our model learns this capability naturally during *pretraining* on a *general corpus*, appending a latent token after every token rather than just at the end of a prompt. PonderLM employs a *vertical* scaling strategy, deepening the model for a single generation step by iteratively re-feeding a "pondering embedding"—a probability-weighted sum of token embeddings—into its input layers. In contrast, our method utilizes a *horizontal* scaling approach. We extend the generative process for each token by appending latent thoughts, which are directly derived from the last hidden state of the previous computation step. A more comprehensive comparison with related works is provided in Table 1.

Other related methods (including test-time scaling and parameter sharing) can be broadly categorized into three main paradigms: scaling model depth via sequential parameter sharing, exploring multiple solutions through parallel computation, and scaling generation steps.

Table 1: A taxonomy of most related methods. The Computation Space column specifies where additional computation occurs. Our method is unique in its ability to learn a per-token, latent-space computational mechanism from a general corpus via a standard pretraining objective, without requiring specialized instruction data or complex training schemes like reinforcement learning.

| Method | Core Strategy | Training Data | Computation Space | Application Level | Training Method |
|---|---|---|---|---|---|
| CoT | Scaling Generation | CoT Data | Explicit Token | Per Question | RL/SFT |
| Pause Tokens | Scaling Generation | General Corpus | Fixed Special Token | Per Token | Pretrain |
| Quiet-STaR | Scaling Generation | General Corpus | Explicit Token | Per Token | RL |
| PonderLM | Scaling Model Depth | General Corpus | Continuous Embedding | Per Token | Pretrain |
| LoopedLM | Scaling Model Depth | General Corpus | Hidden State | Per Token | Pretrain |
| Coconut | Scaling Generation | CoT Data | Hidden State | Per Question | SFT |
| Ours | Scaling Generation | General Corpus | Hidden State | Per Token | Pretrain |

**Sequential Parameter Sharing to Scale Up Model Depth.** This paradigm increases a model's effective depth by reusing its parameters. Early work like Universal Transformers (Dehghani et al.) reused entire blocks, while more recent methods refine this by iterating over layers to refine hidden states (Geiping et al., 2025), recycling output states back into the input (Giannou et al., 2023; Saunshi et al.), or recurrently applying a single layer to critical tokens (Chen et al., 2025a). While these approaches can enhance model capabilities, they often introduce significant inference overhead and training instabilities. In contrast, our horizontal scaling approach avoids the potential instabilities of deep recurrent computations by integrating thought into the sequence length.

**Exploring Multiple Solutions through Parallel Computation.** This paradigm involves generating multiple candidate solutions in parallel and then selecting the most promising one using a specific criterion. Prominent examples include Best-of-N sampling (Cobbe et al., 2021; Sun et al., 2024; Gui et al., 2024; Amini et al., 2024; Sessa et al., 2024) and Majority Voting (Wang et al., 2022). While effective at improving performance on complex reasoning tasks, these approaches can be computationally inefficient. Furthermore, a key challenge is the difficulty of reliably identifying the best candidate from the generated set, as the verifier or selection heuristic may not be optimal (Stroebl et al., 2024; Hassid et al., 2024).

**Scaling Generation Steps.** The most prominent method for scaling generation steps is Chain-of-Thought (CoT) (Wei et al., 2022), which elicits reasoning paths from models before they provide a final answer. While effective, this process is often applied at the per-question level. Subsequent work has sought to integrate this "thinking" process more granularly into generation. One approach involves inserting non-content or "thinking" tokens into the sequence. For example, Goyal et al. (2023) inserted learnable "pause" tokens, while others explored discrete planning tokens (Wang et al., 2024) or filler tokens (Pfau et al., 2024). Quiet-STaR (Zelikman et al., 2024) even uses reinforcement learning to generate explicit rationale tokens between output tokens. However, these methods remain constrained to the discrete vocabulary space. In contrast, our work elevates this per-token computation into the continuous latent space.

## 3 METHODOLOGY

In this section, we introduce our pretraining methodology, which trains a language model to first generate an intermediate latent thought before predicting the actual subsequent token. We first establish the notation for a standard Transformer-based language model. Given an input sequence $x = (x_1, x_2, \ldots, x_T)$, the model processes the token embeddings $\mathbf{E}_t = [\mathbf{e}(x_1), \mathbf{e}(x_2), \ldots, \mathbf{e}(x_t)]$ using a Transformer architecture. The operation can be formulated as:

$$\mathbf{H}_t = \text{Transformer}(\mathbf{E}_t)$$

where $\mathbf{e}(\cdot)$ is the token embedding lookup function. The resulting matrix $\mathbf{H}_t \in \mathbb{R}^{t \times d}$ contains the sequence of last-layer hidden states. We denote the hidden state at position $t$ as $\mathbf{h}_t = \mathbf{H}_t[t, :]$.

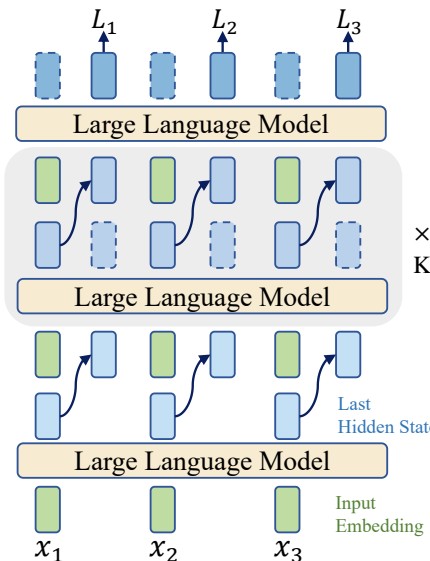

Figure 3: Parallel training procedure of our method (via Jacobi iteration). **(1)** The model computes initial hidden states from the input embeddings $(x_1, x_2, x_3)$. These hidden states are then interleaved with their corresponding token embeddings to form a new input sequence. **(2)** For K rounds, all hidden states are updated in parallel. In each iteration, hidden states from the previous step are interleaved with the original embeddings to form the new input. **(3)** Finally, the cross-entropy loss $(\mathcal{L}_1, \mathcal{L}_2, \mathcal{L}_3)$ is computed at the positions corresponding to the hidden state inputs to optimize language modeling.

## 3.1 INFERENCE PROCESS

The inference process of our model is straightforward (Figure 2). For each token to be generated, the model first computes its corresponding last hidden state. This hidden state is then used as the input embedding for the subsequent token generation step, mimicking a recurrent thinking process.

## 3.2 TRAINING PROCEDURE

While inference is sequential, a purely autoregressive training procedure is computationally infeasible for long sequences (e.g., $T = 2048$), as it would require thousands of separate forward passes. To address this, we employ the **Jacobi iteration** to approximate the true autoregressive hidden states, which allows for parallel training (Figure 3). The goal is to find a set of "fixed-point" hidden states $\mathbf{H}^* = [\mathbf{h}_1^*, \ldots, \mathbf{h}_T^*]$ that are the output of the model when they are also part of the input. We solve for these states iteratively as follows:

**1. Initial Hidden State Estimation (Iteration $0$):** We begin by performing a single forward pass on the original token embeddings $\mathbf{E} = [\mathbf{e}(x_1), \ldots, \mathbf{e}(x_T)]$ to obtain the initial hidden states:

$$[\mathbf{h}_1^0, \mathbf{h}_2^0, \ldots, \mathbf{h}_T^0] = \text{Transformer}([\mathbf{e}(x_1), \mathbf{e}(x_2), \ldots, \mathbf{e}(x_T)])$$

**2. Parallel State Update via Jacobi Iteration (Iteration $k \rightarrow k + 1$):** For each subsequent iteration $k$, we construct a new input sequence by interleaving the original token embeddings with the hidden states from the *previous* iteration, $\mathbf{H}^k$:

$$\mathbf{S}^k = [\mathbf{e}(x_1), \mathbf{h}_1^k, \mathbf{e}(x_2), \mathbf{h}_2^k, \ldots, \mathbf{e}(x_T), \mathbf{h}_T^k]$$

We then feed this sequence into the model to compute the updated hidden states for the next iteration, $\mathbf{H}^{k+1}$, in a single, parallel forward pass:

$$[\ldots, \mathbf{h}_1^{k+1}, \ldots, \mathbf{h}_2^{k+1}, \ldots] = \text{Transformer}(\mathbf{S}^k)$$

In this iterative process, all components of the new state vector $\mathbf{H}^{k+1}$ are computed in parallel based on the entire state vector from the previous iteration, $\mathbf{H}^k$. As shown in Figure 4, this iteration converges rapidly, with the hidden states stabilizing after a few rounds.

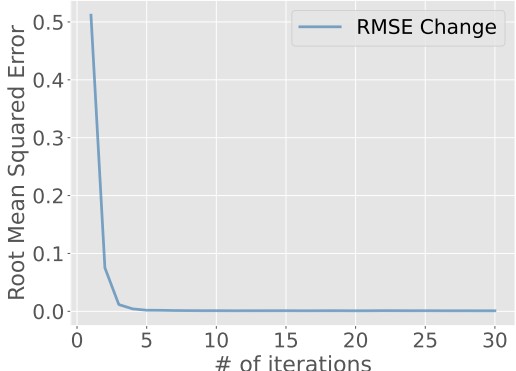

Figure 4: RMSE of the last hidden states before and after the ith iteration. The model is the vanilla Pythia-1B tested with 4*2048 tokens.

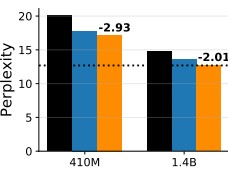 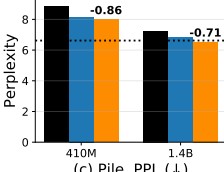 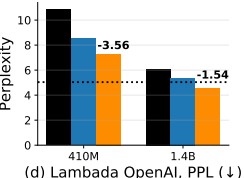

Figure 5: Language Modeling Perplexity (PPL). Our method achieves the lowest perplexity, consistently surpassing PonderLM despite its $2\times$ inference overhead at the same model size. Ours 1.4B (Pythia Arch) also outperforms the larger Pythia-2.8B baseline. Numbers denote the absolute perplexity improvement ($\downarrow$) over the corresponding Pythia models.

**3. Loss Computation:** After $K$ Jacobi iterations, we form the final input sequence $\mathbf{S}^K = [\mathbf{e}(x_1), \mathbf{h}_1^K, \ldots, \mathbf{e}(x_T), \mathbf{h}_T^K]$. The language modeling objective is then optimized by computing the cross-entropy loss $(L_1, L_2, \ldots, L_T)$ at the positions corresponding to the final hidden state inputs. Specifically, the loss $L_i$ is computed for predicting the token $x_{i+1}$ from the hidden state $\mathbf{h}_i^K$. To prevent overfitting to a fixed number of steps, we randomly sample $K$ from $\{2, 3, 4\}$ for each training instance.

By formulating the training in this manner, we break the strict sequential dependency inherent in autoregressive models, thereby enabling efficient, parallel training.

### 3.3 POSITION EMBEDDING

When a last hidden state is fed back into the model as an input, it inherits the same positional encoding as its corresponding original token embedding. For instance, the positional encoding for $\mathbf{h}_i^k$ is identical to that of $\mathbf{e}(x_i)$ for all iterations $k$.

## 4 EXPERIMENTS

Our evaluation comprises six main components:

1. We first present the results of large-scale pretraining for our model on the 300B-token Pile dataset (Gao et al., 2020), analyzing its scaling behavior and language modeling performance in comparison to the official Pythia suite (Biderman et al., 2023).
2. Next, we assess our model on a broad range of downstream tasks, including general benchmarks and instruction-following, and compare its performance with the official Pythia suite, the PonderLM-Pythia (Zeng et al., 2025), and established models such as OPT (Zhang et al., 2022), Bloom (Le Scao et al., 2023), and TinyLLaMA (Zhang et al., 2024).
3. We further benchmark our approach against several competitive baselines, including Looped Transformer (Giannou et al., 2023), Pause Token (Goyal et al., 2023), PonderLM, and models with doubled parameter counts.
4. To validate the effectiveness of our method on off-the-shelf foundation models, we perform continual pretraining on Llama-3-3B (Grattafiori et al., 2024).
5. Finally, we conduct an ablation study to analyze the impact of key hyperparameters.

### 4.1 LARGE-SCALE PRETRAINING ON PILE

We begin by validating our method at scale. We select the Pile (Gao et al., 2020), a substantial 300B-token dataset, as it provides a comprehensive pretraining corpus while remaining computationally tractable.We pretrain models based on the Pythia architecture (Biderman et al., 2023) for two key reasons.. First, its training protocol, including all hyperparameters, is publicly available, enabling a highly controlled experiment where the gains from our method can be isolated. Second, this foundation allows for a direct and rigorous comparison against both the official Pythia models and relevant prior work like PonderLM-Pythia (Zeng et al., 2025).

Table 2: Zero-shot and five-shot accuracy (%) on downstream tasks. All pretrained model weights used for comparison are obtained from their official repositories. Δacc indicates the average accuracy improvement over the corresponding Pythia baseline. Italicized models are shown but not bolded, since they use significantly larger training data or parameters, and their avg acc are marked in red when outperformed by our model. Ponder refers to the PonderLM-Pythia model, whose inference cost is *twice* ours under the same parameter size, with results taken from their original paper.

| Model (#training tokens) | Lambada OpenAI | ARC -E | Lambada Standard | ARC -C | Wino Grande | PIQA | Hella Swag | SciQ | RACE | Avg acc / Δacc ↑ |
|---|---|---|---|---|---|---|---|---|---|---|
| **0-shot** | | | | | | | | | | |
| Pythia-410M (300B) | 51.4 | 52.2 | 36.4 | 21.4 | 53.8 | 66.9 | 33.7 | 81.5 | 30.9 | 47.6 |
| OPT-350M (300B) | 45.2 | 44.0 | 35.8 | 20.7 | 52.3 | 64.5 | 32.0 | 74.9 | 29.8 | 44.4 |
| Bloom-560M (366B) | 34.3 | 47.5 | 33.3 | 22.4 | 51.5 | 63.8 | 31.5 | 80.3 | 30.5 | 43.9 |
| Ponder-410M (300B) | 56.9 | 51.9 | 45.3 | 22.6 | **56.0** | 68.7 | 37.0 | 81.4 | **33.8** | 50.4 |
| *Pythia-1B (300B)* | *55.9* | *56.8* | *42.0* | *24.2* | *52.5* | *70.5* | *37.7* | *83.3* | *32.7* | *50.6* |
| **Ours-410M** (Pythia Arch, 300B) | **59.1** | **54.0** | **47.3** | **24.6** | 55.5 | 69.4 | 37.7 | **86.2** | 33.5 | **51.9** / +4.3 |
| Pythia-1.4B (300B) | 61.6 | 60.4 | 49.7 | 25.9 | 57.5 | 70.8 | 40.4 | 86.4 | 34.1 | 54.1 |
| OPT-1.3B (300B) | 57.9 | 57.1 | 52.5 | 23.4 | 59.7 | 71.8 | 41.6 | 84.3 | 34.3 | 53.6 |
| Bloom-1.7B (366B) | 46.2 | 56.4 | 44.5 | 23.7 | 56.8 | 68.5 | 37.5 | 85.0 | 33.2 | 50.2 |
| Ponder-1.4B (300B) | 65.2 | 62.0 | 53.8 | 27.0 | 60.1 | 72.6 | 44.0 | 89.0 | 35.2 | 56.5 |
| *Tinyllama-1.1B (3T)* | *58.8* | *60.3* | *49.3* | *28.0* | *59.0* | *73.3* | *45.0* | *88.9* | *36.4* | *55.4* |
| *Pythia-2.8B (300B)* | *64.6* | *64.4* | *54.3* | *29.5* | *60.2* | *73.8* | *45.4* | *88.5* | *34.9* | *57.3* |
| **Ours-1.4B** (Pythia Arch, 300B) | **67.6** | **64.1** | **57.1** | **30.2** | **60.9** | 72.9 | **45.8** | **91.0** | **37.1** | **58.5** / +4.4 |
| **5-shot** | | | | | | | | | | |
| Pythia-410M (300B) | 43.9 | 54.7 | 32.8 | 22.3 | 53.4 | 68.0 | 33.8 | 88.9 | 30.4 | 47.6 |
| OPT-350M (300B) | 38.3 | 45.4 | 32.1 | 20.5 | 53.0 | 65.8 | 31.9 | 85.7 | 29.5 | 44.7 |
| Bloom-560M (366B) | 29.4 | 50.2 | 29.7 | 21.9 | 52.7 | 64.2 | 31.4 | 88.0 | 30.0 | 44.2 |
| Ponder-410M (300B) | 48.9 | **58.7** | 43.7 | **26.1** | 54.0 | 70.5 | 37.3 | 91.0 | 32.4 | 51.4 |
| *Pythia-1B (300B)* | *48.3* | *58.6* | *35.8* | *25.4* | *52.8* | *71.3* | *37.7* | *91.6* | *31.7* | *50.4* |
| **Ours-410M** (Pythia Arch, 300B) | **52.1** | 58.0 | **45.0** | 26.0 | **54.6** | 69.2 | 37.9 | 91.7 | 32.6 | **51.9** / +4.3 |
| Pythia-1.4B (300B) | 54.5 | 63.1 | 44.5 | 28.8 | 57.1 | 71.0 | 40.5 | 92.4 | 34.6 | 54.1 |
| OPT-1.3B (300B) | 54.0 | 60.4 | 49.0 | 26.9 | 56.9 | 72.4 | 38.5 | 91.8 | 35.4 | 52.7 |
| Bloom-1.7B (366B) | 42.5 | 58.8 | 41.5 | 26.2 | 57.7 | 68.7 | 37.6 | 91.9 | 33.5 | 50.9 |
| Ponder-1.4B (300B) | 59.2 | **67.5** | 49.9 | 32.4 | 60.4 | 73.5 | 44.2 | 94.3 | 37.1 | 57.6 |
| *Tinyllama-1.1B (3T)* | *53.8* | *64.8* | *45.0* | *31.1* | *59.4* | *73.8* | *44.9* | *94.0* | *36.4* | *55.9* |
| *Pythia-2.8B (300B)* | *59.0* | *67.0* | *50.7* | *31.0* | *61.1* | *74.4* | *45.3* | *93.7* | *35.9* | *57.6* |
| **Ours-1.4B** (Pythia Arch, 300B) | **63.6** | 67.4 | **56.0** | **32.6** | **64.0** | 73.5 | **46.4** | **94.5** | **37.9** | **59.5** / +5.4 |

### 4.1.1 SCALING PROPERTIES

As illustrated in Figure 1, our pretrained models demonstrate superior scaling properties in both parameter and data efficiency. Ours-1.26B (Pythia Arch), for instance, matches the performance of the official Pythia-2.8B with 55% fewer parameters. Furthermore, ours-1.4B (Pythia Arch) converges to the official version's final performance using 62% less training data. Additional scaling curves on GPT-2 and LLaMA, presented in Appendix E, further validate the generalizability of our method.

### 4.1.2 LANGUAGE MODELING ABILITY

To further quantify these pretraining gains, we evaluate perplexity (PPL) on several standard benchmarks (Pile validation, Wikitext (Merity et al., 2016), and the Lambada (Paperno et al., 2016)). The results in Figure 5 show that our method delivers substantial and consistent PPL reductions across all model sizes and datasets. Notably, ours-1.4B (Pythia Arch) is better than offical Pythia-2.8B.

## 4.2 DOWNSTREAM TASK EVALUATION

We now evaluate the practical capabilities of our previous pretrained Pythia models on a range of downstream applications.

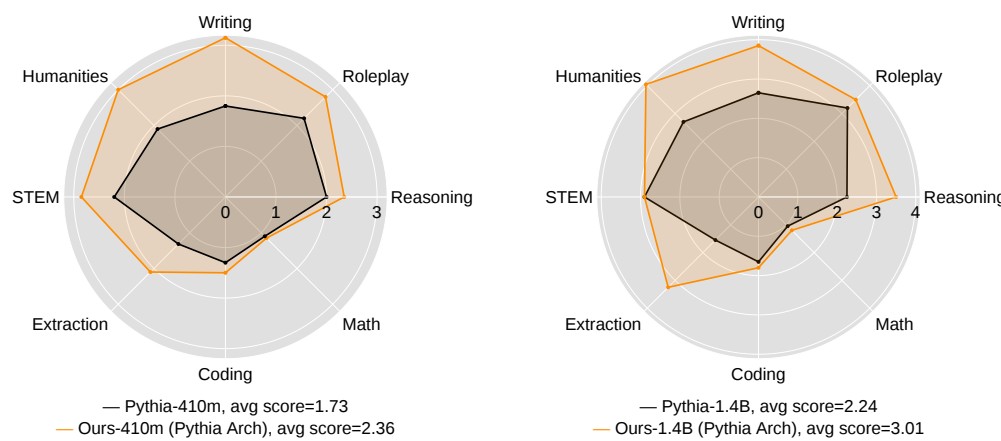

Figure 6: Instruction-following evaluation on MT-Bench. Our pretrained Pythia models outperform their official Pythia counterparts in all categories.

### 4.2.1 GENERAL DOWNSTREAM TASKS

**Datasets.** Following (Gu & Dao, 2023; Zeng et al., 2025), we including LAMBADA (Paperno et al., 2016), SciQ (Welbl et al., 2017), HellaSwag (Zellers et al., 2019), PIQA (Bisk et al., 2020), WinoGrande (Sakaguchi et al., 2021), ARC-Easy and ARC-Challenge (Clark et al., 2018), RACE (Lai et al., 2017) for comprehensive evaluation.

**Baselines.** We compare our pretrained Pythia models against several strong baselines: (1) the official Pythia models; (2) the PonderLM-Pythia models from prior work; (3) several open-source models trained on a similar data volume, including OPT (Zhang et al., 2022) and Bloom (Le Scao et al., 2023); and (4) TinyLLaMA (Zhang et al., 2024), a powerful model trained on ten times the amount of data (3T tokens vs. our 300B).

**Results.** As shown in Table 2, our pretrained Pythia models consistently outperform similarly-sized baselines, including official Pythia, PonderLM-Pythia, OPT, and Bloom. Remarkably, our models also surpass competitors more than twice their size. For instance, ours-410M (Pythia Arch) exceeds the performance of official Pythia-1B and Bloom-1.7B, while ours-1.4B (Pythia Arch) outperforms Pythia-2.8B. Furthermore, our pretrained Pythia-1.4B significantly surpasses TinyLLaMA-1.1B, despite the latter being trained on $10\times$ more data.

### 4.2.2 INSTRUCTION-FOLLOWING ABILITY

We evaluate instruction-following capabilities by fine-tuning the 410m and 1.4B versions of ours (Pythia Arch) and the official Pythia models on the Alpaca dataset (Taori et al., 2023). When tested on the MT-Bench benchmark (Zheng et al., 2023), the our pretrained Pythia models consistently outperform their official Pythia counterparts across all categories, as shown in Figure 6. This results in significant average score improvements of 0.63 for the 410m model and 0.77 for the 1.4B model.

### 4.3 COMPARISON WITH BASELINE METHODS

To contextualize the performance and efficiency of our proposed method, we conduct a detailed comparison against several competitive baselines on the LLaMA architecture.

**Baselines.** We compare our model against four strong approaches:

- **Looped Transformer** (Saunshi et al.): Processes the input by iterating through the entire set of transformer layers multiple times.
- **Pause Token** (Goyal et al., 2023): Inserts a specified number of learnable "pause tokens" before generating each token, allowing for additional computation per step.

Table 3: Comparison on various benchmarks. Inference FLOPs are relative to the vanilla model. Actual throughput is evaluated following Wu & Tu (2024). Our method shows superior performance across all metrics while maintaining inference efficiency. Detailed downstream tasks performance are provided in Appendix D.

| Model | Inference FLOPs | Throughput (tokens/s) | Pile | Lambda OpenAI | Wikitext | Lambda Standard | Avg Acc 0 shot | Avg Acc 5 shot |
|---|---|---|---|---|---|---|---|---|
| LLaMA-1.4B (train from scratch) | 1× | 221.19 | 9.04 | 11.42 | 20.18 | 27.52 | 47.7 | 47.5 |
| **Methods with comparable (2×) inference FLOPs** | | | | | | | | |
| Looped LLaMA-1.4B (2 loops) | 2× | 112.33 | 8.35 | 9.22 | 18.34 | 21.50 | 49.8 | 48.5 |
| Pause LLaMA-1.4B (1 pause) | 2× | 112.57 | 8.58 | 9.87 | 19.10 | 19.90 | 48.5 | 48.2 |
| Pondering LLaMA-1.4B (1 step) | 2× | 110.16 | 8.33 | 9.26 | 18.36 | 19.95 | 49.6 | 49.2 |
| LLaMA-2.8B (train from scratch) | 2× | 110.50 | 8.23 | 8.93 | 18.09 | 17.08 | 49.8 | 50.6 |
| **Our LLaMA-1.4B** | 2× | 111.55 | **7.89** | **7.39** | **16.99** | **12.20** | **52.3** | **51.9** |
| **Methods with higher (4×) inference FLOPs** | | | | | | | | |
| Looped LLaMA-1.4B (4 loops) | 4× | 59.48 | 8.04 | 8.14 | 17.35 | 15.14 | 50.9 | 50.5 |
| Pause LLaMA-1.4B (3 pauses) | 4× | 60.43 | 8.17 | 7.92 | 17.81 | 13.76 | 51.0 | 50.4 |
| Pondering LLaMA-1.4B (3 steps) | 4× | 55.42 | 8.03 | 8.02 | 17.23 | 15.48 | 51.5 | 51.5 |

- **Pondering LLM** (Zeng et al., 2025): Iteratively refines its output by feeding a "pondering embedding" (a probability-weighted sum of token embeddings) back into the model for several steps.
- **Scaled-up Model**: As an oracle baseline, we train a standard LLaMA model with twice the number of parameters (2.8B). This model has inference FLOPs comparable to our method.

For the iterative baselines (Looped Transformer, Pause Token, and PonderLM), we evaluate them under two distinct computational budgets. First, we configure them to match the inference FLOPs of our method (a 2× increase over the vanilla model), corresponding to 2 loops, 1 pause token, or 1 pondering step. Second, to provide a more challenging comparison, we pretrain them in a setting with double the inference FLOPs of our method (a 4× increase), corresponding to 4 loops, 3 pause tokens, or 3 pondering steps.

**Settings.** We use the LLaMA-1.4B model as our testbed. For a fair comparison, all models are trained on a 26B token dataset using identical hyperparameters. We report perplexity (PPL; lower is better) on the Pile validation set, Wikitext, and the Lambda datasets (OpenAI and standard versions), as well as the average accuracy on the nine downstream tasks previously mentioned. The computational overhead is measured in relative inference FLOPs against the vanilla 1.4B model.

**Training Computation Analysis.** We analyze the training FLOPs relative to the vanilla model when trained on the same amount of data. For the baselines configured with the higher (4×) inference budget (e.g., 4 loops, 3 pause tokens, or 3 pondering steps), the training cost scales linearly, resulting in approximately 4× the FLOPs of the vanilla baseline. The scaled-up model (2.8B) incurs roughly 2× the training FLOPs due to the doubled parameter count.

For our method, the training process involves three components: (1) an initial forward pass on the original sequence (1×); (2) $K$ Jacobi iterations performed on an interleaved sequence of tokens and thoughts (which doubles the sequence length, incurring 2× cost per iteration); and (3) a final forward pass on the interleaved sequence (2×). The total training cost multiplier is therefore formulated as $1 + 2K + 2 = 3 + 2K$. In our experiments, we sample $K$ from $\{2, 3\}$ ($\mathbb{E}[K] = 2.5$) rather than $\{2, 3, 4\}$ to prioritize efficiency as we empirically found this range sufficient, resulting in an average training cost of approximately $3 + 2 \times 2.5 = 8\times$ that of the vanilla baseline.

**Results.** The results, summarized in Table 3, show that our method consistently achieves the lowest perplexity across all language modeling benchmarks and the highest average accuracy on the downstream tasks. Notably, our approach not only surpasses all methods in the comparable (2×) inference FLOPs category, including the LLaMA-2.8B oracle, but also demonstrates a significant advantage over methods operating at a much higher (4×) inference budget. This highlights the superior performance and inference efficiency of our approach.

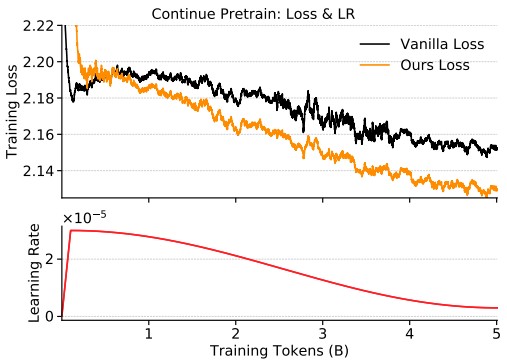 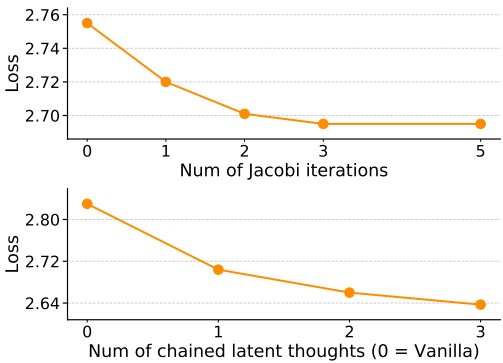

Figure 7: Training loss comparison of our method and vanilla continual pretraining for the LLaMA-3-3B on 5B tokens from SlimPajama.

Figure 8: (Top) Impact of the num of Jacobi iterations . (Bottom) Chaining more latent thoughts before each token consistently lowers loss.

Table 4: 0-shot and 5-shot accuracy (%) on downstream tasks, evaluating LLaMA-3-3B enhancement via continual pre-training (CPT). The table compares three models: original LLaMA-3-3B, vanilla CPT, and our method with continual pre-training.

| Model | Lambada OpenAI | ARC -E | Lambada Standard | ARC -C | Wino Grande | PIQA | Hella Swag | SciQ | RACE | Avg acc / $\Delta$acc ↑ |
|---|---|---|---|---|---|---|---|---|---|---|
| **0-shot** | | | | | | | | | | |
| LLaMA-3-3B | 70.1 | 74.5 | 63.7 | 42.2 | 69.0 | 76.8 | 55.4 | 95.5 | 39.4 | 65.2 |
| Standard CPT | 69.4-0.7 | 76.2+1.7 | 65.1+1.4 | 42.5+0.3 | 67.5-1.5 | 77.2+0.4 | 55.1-0.3 | 94.1-1.4 | 39.6+0.2 | 65.2 |
| **Ours CPT** | 70.8+0.7 | 76.3+1.8 | 67.5+3.8 | 42.1-0.1 | 69.0+0.0 | 77.9+1.1 | 56.1+0.7 | 94.6-0.9 | 40.5+1.1 | **66.2** |
| **5-shot** | | | | | | | | | | |
| LLaMA-3-3B | 66.8 | 78.1 | 64.1 | 44.1 | 71.4 | 78.6 | 56.1 | 96.4 | 41.8 | 66.4 |
| Standard CPT | 66.1-0.7 | 77.7-0.4 | 66.0+1.9 | 43.0-1.1 | 71.5+0.1 | 78.3-0.3 | 55.8-0.3 | 96.5+0.1 | 40.6-1.2 | 66.2 |
| **Ours CPT** | 69.0+2.2 | 78.6+0.5 | 67.4+3.3 | 48.0+3.9 | 72.9+1.5 | 78.1-0.5 | 57.0+0.9 | 96.7+0.3 | 41.4-0.4 | **67.7** |

## 4.4 Effectiveness on off-the-shelf Foundation Models

We further investigate whether our method can effectively enhance existing, large-scale foundation models. To this end, we take the official LLaMA-3-3B (Grattafiori et al., 2024) model and perform continual pre-training on 5 billion tokens from the SlimPajama dataset (Shen et al., 2024). We compare the original model's performance with two versions after this additional training: a vanilla continual pretraining baseline and our proposed approach. As illustrated in Figure 7, our method achieves a lower training loss than the vanilla baseline after consuming less than 1B tokens, and this performance gap widens as training progresses. As shown in Table 4, evaluation on our nine standard downstream tasks reveals that our method provides a substantial boost in performance, demonstrating its utility as a plug-and-play enhancement for off-the-shelf models.

## 4.5 Ablation Study

In this section, we study the impact of key components. We ablate the number of Jacobi iterations, different position embedding strategies, and the number of latent thoughts (chain analogous to CoT) generated before each token. All experiments are conducted on the Pythia-70m with 30B tokens.

As shown in Figure 8 (Top), increasing the number of Jacobi iterations initially lowers the loss, but the improvement saturates after only 3 iterations, which corroborates the fast convergence we observe (Figure 4). Meanwhile, as illustrated in Figure 8 (Bottom), chaining more latent thoughts consistently leads to better performance, which demonstrates our method's potential for pretraining LLMs to generate a chain of latent thoughts before predicting each token. Regarding position embedding, we compared assigning sequential position ids to thoughts versus reusing the token's

position id for its corresponding thoughts and found a negligible performance difference. We use the latter strategy to avoid the need for long-context capabilities.

## 5 CONCLUSION

In this paper, we introduce the approach of pretraining language models with latent thoughts, which can be effectively realized using large-scale general corpora. Language models pretrained with latent thoughts consistently outperforms its counterparts with double the parameters (at equal inference cost), as well as prior related methods like PonderLM, looped, and paused models, even when they use double the inference budget. Furthermore, we show that chaining latent thoughts, akin to COT, consistently improves model performance. We posit that our work introduces a new potential dimension for scaling the capabilities of language models.

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

## A  ETHICS STATEMENT

Our model's demonstrated superiority in performance and efficiency necessitates a discussion of its dual-use nature. The same advanced capabilities that make it a powerful tool for positive applications could also be leveraged to generate highly convincing misinformation at scale. Furthermore, its novel architecture may present unknown security and privacy risks. We believe the responsible advancement of this technology requires a parallel effort to understand and mitigate these challenges.

## B  REPRODUCIBILITY STATEMENT

To ensure the reproducibility of our research, we have included the complete source code in the supplementary material. We also provide core hyperparameter and implementation settings in this paper. Using the provided code and specified configurations, all main results and figures reported in this paper can be fully reproduced. A public code repository will be made available upon publication.

## C  THE USE OF LARGE LANGUAGE MODELS

Large Language Models (LLMs) were not used for the core methodology or the main research content of this paper. We utilized an LLM solely for the purpose of improving the language and clarity of the manuscript.

## D  DETAILED DOWNSTREAM TASKS PERFORMANCE

Table 5: Zero-shot and five-shot accuracy (%) on downstream tasks, as described in Section 4.3.

| Model | Lambada OpenAI | ARC -E | Lambada Standard | ARC -C | Wino Grande | PIQA | Hella Swag | SciQ | RACE | Avg acc / $\Delta$acc $\uparrow$ |
|---|---|---|---|---|---|---|---|---|---|---|
| **0-shot** | | | | | | | | | | |
| LLaMA-1.4B (train from scratch) | 50.6 | 52.2 | 37.1 | 20.9 | 53.0 | 65.6 | 33.6 | 84.5 | 31.8 | 47.7 |
| **Methods with comparable ($2\times$) inference FLOPs** | | | | | | | | | | |
| Looped LLaMA-1.4B (2 loops) | 53.6 | 54.3 | 41.8 | 23.6 | 52.9 | 69.3 | 35.8 | 83.6 | **33.5** | 49.8 |
| Pause LLaMA-1.4B (1 pause) | 51.8 | 53.8 | 40.1 | 21.8 | 52.5 | 67.5 | 34.7 | 83.1 | 30.9 | 48.5 |
| Pondering LLaMA-1.4B (1 step) | 53.8 | 53.2 | 42.6 | 23.0 | 52.6 | 68.4 | 35.9 | 83.2 | 33.3 | 49.6 |
| LLaMA-2.8B (train from scratch) | 54.3 | 53.4 | 43.5 | 24.4 | 53.1 | 68.0 | 36.2 | 83.4 | 31.5 | 49.8 |
| **Our LLaMA-1.4B** | **58.1** | **58.0** | **48.2** | **25.2** | **53.9** | **70.7** | **38.6** | **85.9** | 32.4 | **52.3/+4.6** |
| **Methods with higher ($4\times$) inference FLOPs** | | | | | | | | | | |
| Looped LLaMA-1.4B (4 loops) | 55.8 | 55.2 | 45.6 | 23.2 | 54.1 | 68.9 | 37.6 | 84.9 | 33.0 | 50.9 |
| Pause LLaMA-1.4B (3 pauses) | 56.2 | 54.0 | 46.5 | 24.2 | 55.3 | 68.8 | 36.7 | 85.4 | 32.3 | 51.0 |
| Pondering LLaMA-1.4B (3 step) | 56.7 | 56.3 | 45.4 | 23.8 | 55.6 | 68.3 | 37.8 | 86.3 | 33.0 | 51.5 |
| **5-shot** | | | | | | | | | | |
| LLaMA-1.4B (train from scratch) | 45.1 | 53.5 | 34.7 | 22.3 | 50.9 | 66.1 | 33.6 | 89.3 | **31.6** | 47.5 |
| **Methods with comparable ($2\times$) inference FLOPs** | | | | | | | | | | |
| Looped LLaMA-1.4B (2 loops) | 46.0 | 55.6 | 36.9 | 23.9 | 51.1 | 69.2 | 35.7 | 90.1 | 27.9 | 48.5 |
| Pause LLaMA-1.4B (1 pause) | 45.5 | 56.1 | 35.6 | 23.8 | 50.8 | 68.0 | 34.9 | 88.4 | 30.7 | 48.2 |
| Pondering LLaMA-1.4B (1 step) | 48.0 | 56.3 | 40.9 | 24.4 | 53.3 | 69.2 | 36.2 | 89.5 | 25.0 | 49.2 |
| LLaMA-2.8B (train from scratch) | 49.7 | 57.2 | 43.7 | 25.3 | 53.1 | 69.3 | 36.3 | 89.6 | 30.8 | 50.6 |
| **Our LLaMA-1.4B** | **49.8** | **59.6** | **45.6** | **27.7** | **56.3** | **69.8** | **38.7** | **91.3** | 28.0 | **51.9 /+4.4** |
| **Methods with higher ($4\times$) inference FLOPs** | | | | | | | | | | |
| Looped LLaMA-1.4B (4 loops) | 48.0 | 58.5 | 42.8 | 25.0 | 54.8 | 70.4 | 37.6 | 89.2 | 28.5 | 50.5 |
| Pause LLaMA-1.4B (3 pauses) | 48.8 | 56.6 | 41.0 | 24.1 | 54.1 | 69.3 | 36.6 | 90.7 | 32.6 | 50.4 |
| Pondering LLaMA-1.4B (3 step) | 49.5 | 58.8 | 42.6 | 25.4 | 54.3 | 69.2 | 37.9 | 91.2 | 34.7 | 51.5 |

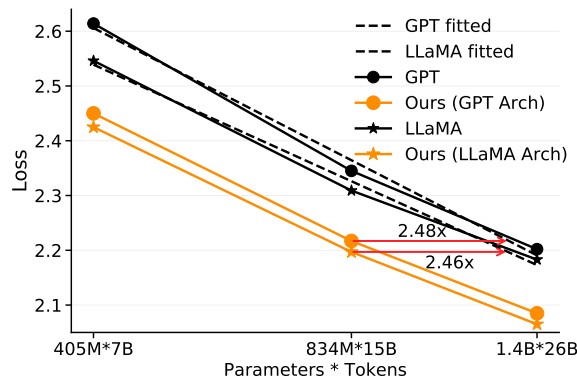

Figure 9: Scaling curves for vanilla models and ours (GPT-2 Arch and LLaMA Arch).

## E    GENERALIZATION TO GPT-2 AND LLaMA

To verify the general applicability of our method, we apply our proposed latent mechanism to the widely-used GPT-2 and LLaMA architectures.

**Experimental Settings.** We train both vanilla and latent-enhanced versions of these models from scratch on a subset of the Pile dataset, with sizes ranging from 405M to 1.4B parameters. The experimental setup, including the number of training tokens aligned with Chinchilla scaling laws (Hoffmann et al., 2022b). Detailed model configurations and training hyperparameters are provided in Table 6.

**Results.** The scaling curves are presented in Figure 9. Our method provides significant and consistent performance improvements for both GPT-2 and LLaMA across all model sizes. Notably, ours-834M (GPT-2 Arch) and ours-834M (LLaMA Arch) achieve a validation loss comparable to their vanilla counterparts trained with approximately **2.48x** and **2.46x** the parameter-token product, respectively.

Table 6: Model sizes and hyperparameters for the scaling experiments on GPT-2 and LLaMA.

| Parameters | $n_{layers}$ | $d_{model}$ | $n_{heads}$ | Learning Rate | Batch Size (tokens) | Training Tokens |
|---|---|---|---|---|---|---|
| 405M | 24 | 1024 | 16 | 3.0e-4 | 0.5M | 7B |
| 834M | 24 | 1536 | 24 | 2.5e-4 | 0.5M | 15B |
| 1.4B | 24 | 2048 | 32 | 2.0e-4 | 0.5M | 26B |

## F    COMPLEMENTING MODELS WITH TEST-TIME SCALING APPROACHES

We study whether our method (denoted **Ours CPT**) complement common test-time scaling strategies, compared with a vanilla baseline (**Standard CPT**). We evaluate on TRUTHFULQA (ROUGE-L) and GSM8K (Exact Match), and set the number of samples $N \in \{1, \ldots, 10\}$ for Best-of-$N$ and Majority Voting.

**Settings.** We reuse the two models from Section 4.4: both start from the official LLaMA-3-3B backbone and undergo continual pretraining on the same SlimPajama dataset. One is a vanilla continual-pretraining baseline (Standard CPT), while the other incorporates our latent thoughts during continual pretraining (Ours CPT). We evaluate three test-time strategies: Majority Voting, Best-of-$N$ (BoN), and Chain-of-Thought (CoT) prompting, varying only the number of samples $N$ for voting and BoN.

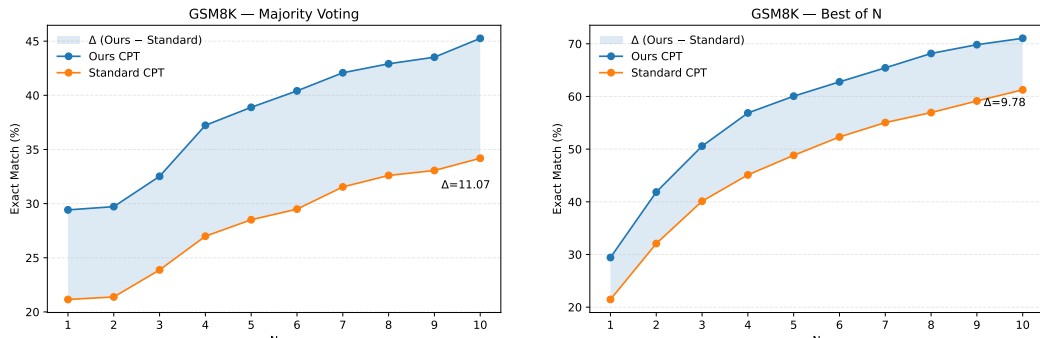

Figure 10: Test-time scaling on GSM8K. Majority Voting (left) and Best-of-$N$ (right). Across $N \in \{1, \ldots, 10\}$, Ours CPT improves more than the baseline.

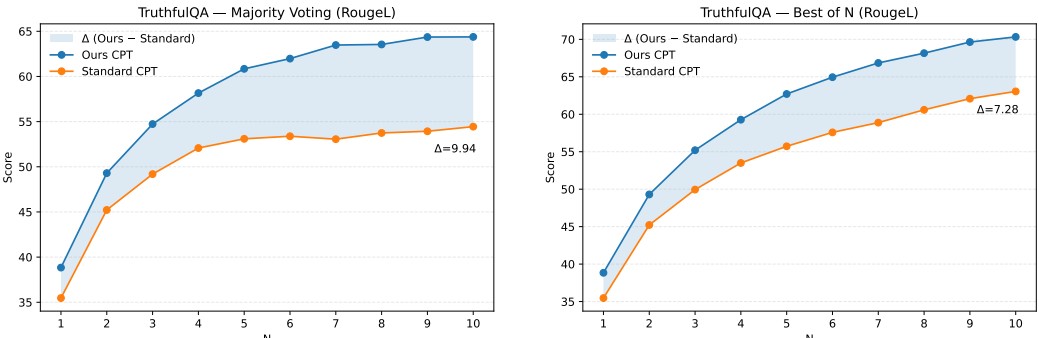

Figure 11: Test-time scaling on TRUTHFULQA. Majority Voting (left) and Best-of-$N$ (right). Ours CPT benefits more than Standard CPT across $N$.

## F.1 MAJORITY VOTING

As $N$ increases, both models improve, but the gap between them widens consistently on TRUTH-FULQA and GSM8K. This pattern suggests that Ours CPT produces batches of answers that are clustered and consistently correct, making aggregation particularly effective. (See Figure 10 (left) and Figure 11 (left).)

## F.2 BEST OF N

BoN likewise amplifies Ours CPT over Standard CPT and improves monotonically with $N$, indicating that our model reliably produces a *diverse set of high-quality candidates* from which a strong single answer can be selected. While its gains are typically a bit smaller than Majority Voting at the same $N$, the strong BoN curve is nevertheless evidence against mode collapse: Ours CPT generates multiple plausible solutions, and either selecting the best (BoN) or aggregating them (voting) yields consistent benefits. (See Figure 10 (right) and Figure 11 (right).)

## F.3 CoT

We further test CoT prompting on GSM8K. As summarized in Table 7, CoT improves both models, with Ours CPT benefiting more. This suggests a complementary relationship between the two techniques, where their combination leads to more reliable outcomes.

Table 7: Effect of CoT on GSM8K (Exact Match; higher is better). CoT improves both models and yields consistently larger gains for Ours CPT.

| Setting | Standard CPT | Ours CPT |
|---|---|---|
| without CoT | 0.1001 | **0.1259** |
| with CoT | 0.2426 | **0.3290** |

# G   JACOBI CONVERGENCE ANALYSIS AND EQUIVALENCE TO SEQUENTIAL INFERENCE

In this section, we provide a rigorous theoretical and empirical analysis to demonstrate that the parallel Jacobi iteration employed during training is mathematically consistent with standard sequential inference. We structure our analysis along a four-step logical chain:

1. **Convergence:** The parallel iteration is theoretically guaranteed to converge to a fixed point $H^*$;
2. **Exponential Rate:** The rapid decay of iterative updates proves that the convergence is exponential;
3. **Trajectory Alignment:** Empirical evidence shows the iteration also converges to the sequential solution $H_{\text{seq}}$;
4. **Equivalence:** By the uniqueness of limits, it follows that $H^* = H_{\text{seq}}$, proving the solutions are identical.

## G.1   EXACT CONVERGENCE GUARANTEE OF PARALLEL TRAINING (FINITE-STEP PROPERTY)

We view the Jacobi iteration in parallel training as a process of finding a fixed point. Let $E \in \mathbb{R}^{T \times d}$ be the fixed input Embeddings, and $H^{(k)} \in \mathbb{R}^{T \times d}$ be the hidden states at iteration $k$. We treat the Transformer layers as a non-linear operator $\Phi$, with the update rule $H^{(k+1)} = \Phi(H^{(k)}; E)$. Our goal is to find the fixed point $H^*$ satisfying $H^* = \Phi(H^*; E)$.

Unlike general fixed-point problems, the Autoregressive Causality of the Transformer guarantees stability. Specifically, since the computation of the $i$-th token depends strictly on preceding tokens ($j < i$), convergence follows a clear inductive chain: the first token stabilizes immediately based on the fixed input, and subsequently, any token $k$ stabilizes once its preceding context (tokens 1 to $k-1$) is fixed.

This guarantees that for a sequence of length $T$, the entire sequence strictly converges to the fixed point $H^*$ in at most $T$ steps.

## G.2   FAST EXPONENTIAL CONVERGENCE

While the above theorem provides a "worst-case" guarantee (taking $T$ steps), the core advantage of our method is that its convergence speed is significantly faster than $T$. We introduce the Banach Fixed-Point Theorem for analysis.

**Theoretical Analysis:**

- **Definition:** According to the Banach Fixed-Point Theorem, if there exists a Lipschitz constant $L$ ($0 \le L < 1$) under some norm $\|\cdot\|$, such that for any $H_a, H_b$, $\|\Phi(H_a) - \Phi(H_b)\| \le L\|H_a - H_b\|$ holds, then the iteration must converge to a unique fixed point $H^*$.
- **Convergence Speed:** If $L < 1$, the error at iteration $k$ will decay exponentially: $\|H^{(k)} - H^*\| \le L^k\|H^{(0)} - H^*\|$. This indicates that the algorithm possesses an exponential convergence property, meaning the error magnitude shrinks by a fixed ratio $L$ at each step.
- **Posterior Error Estimation (Cauchy Property):** Since the true fixed point $H^*$ is unknown, we rely on the property of Cauchy Sequences. The upper bound of the error to the true solution is determined by the RMSE of adjacent iterations:

$$\|H^{(k)} - H^*\| \le \frac{L}{1-L} \underbrace{\|H^{(k)} - H^{(k-1)}\|}_{\text{RMSE } r_k}$$

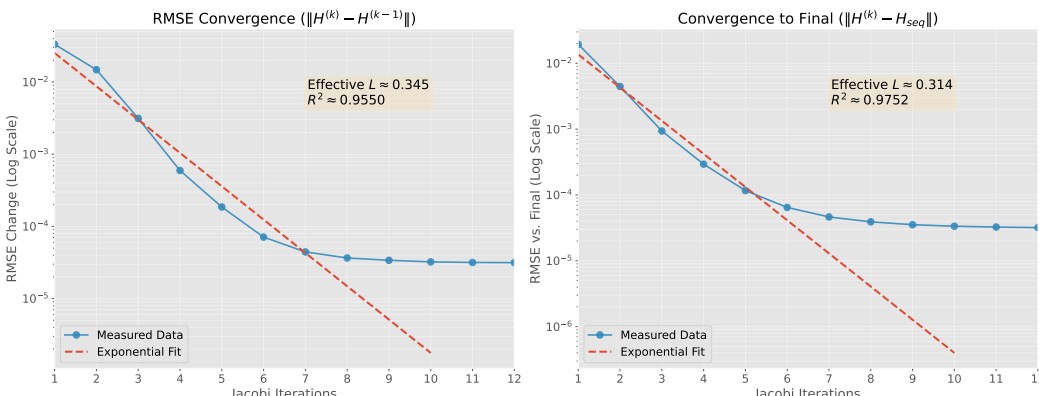

Figure 12: Empirical convergence analysis of the Jacobi iteration on our pretrained Pythia-410M. (Left) The decay of the internal RMSE across 30 iterations shows that the parallel hidden states converge exponentially to a fixed point, eventually hitting the BFloat16 numerical precision floor. (Right) The convergence of the parallel hidden states towards the sequential autoregressive solution over the first 10 iterations. We focus on the initial phase to highlight the rapid exponential alignment with the sequential result before numerical noise dominates.

This inequality implies via a logical chain: as long as we observe the RMSE $r_k$ decaying at rate $L$, it is mathematically guaranteed that the true error $\|H^{(k)} - H^*\|$ also tends to 0 at the same rate $L$. If the operator $\Phi$ is a contraction mapping, then for any adjacent iteration steps:

$$\underbrace{\|H^{(k+1)} - H^{(k)}\|}_{r_{k+1}} = \|\Phi(H^{(k)}) - \Phi(H^{(k-1)})\| \leq L \cdot \underbrace{\|H^{(k)} - H^{(k-1)}\|}_{r_k}$$

Recursively, we obtain $r_k \leq L^k \cdot r_0$. This proves that: under a contraction mapping ($L < 1$), the RMSE $r_k$ must decay exponentially.

**Empirical Verification:** We tracked the RMSE $r_k$ for our pretrained Pythia-410M model. The results are shown in Figure 12 (Left).

- Log-Linear Fitting: To quantify the convergence rate and verify the exponential decay hypothesis ($r_k \propto L^k$), we linearize the relationship by taking the base-10 logarithm:

$$\log_{10}(r_k) \approx k \cdot \log_{10}(L) + C$$

This equation indicates that if the convergence is exponential, the RMSE curve should form a straight line on a semi-logarithmic scale, with the slope corresponding to $\log_{10}(L)$.
- Exponential Decay: In the active convergence phase (first $\sim 10$ iterations), the RMSE exhibits a strict linear trajectory on the semi-logarithmic scale ($R^2 > 0.95$), confirming the exponential decay law consistent with the contraction mapping theory.
- Effective Lipschitz Constant ($L$): The fitting yields $L \approx 0.345 < 1$.
- Practical Implication: $L \approx 0.345$ implies an extremely rapid convergence rate.
  - Reducing the error to 1% ($10^{-2}$) of the initial value requires only $\sim 4$ iterations.
  - Reducing the error to 0.1% ($10^{-3}$) requires only $\sim 6$ iterations.
- Precision Floor: The convergence stabilizes at the BFloat16 precision limit ($\sim 10^{-5}$), confirming the algorithm has reached the maximum precision allowed by the hardware.

### G.3 ITERATION ALSO CONVERGES TO THE SEQUENTIAL SOLUTION $H_{\text{SEQ}}$

"Sequential Inference" produces the standard autoregressive result $H_{\text{seq}}$, and the result of the parallel iteration $H^{(k)}$ also converges to $H_{\text{seq}}$.

**Verification:** To empirically verify this, we calculated the RMSE between the parallel state $H^{(k)}$ at iteration $k$ and the sequential ground truth $H_{\text{seq}}$. As shown in Figure 12 (Right), the distance $\|H^{(k)} - H_{\text{seq}}\|$ similarly exhibits exponential decay. Within $\sim 9$ iterations, the difference drops to the BFloat16 precision floor ($\sim 10^{-5}$).

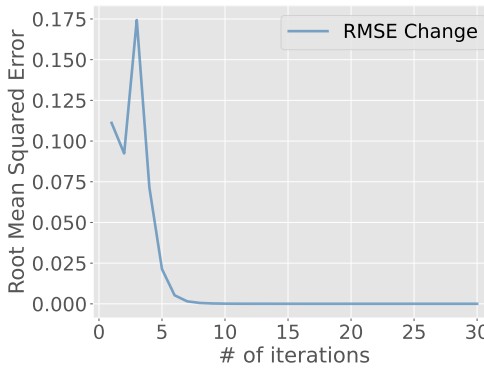 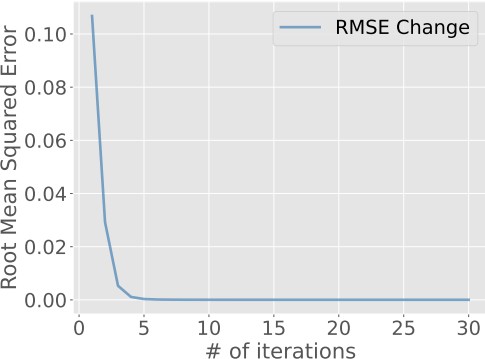

Figure 13: **Randomizing the iteration count prevents depth overfitting.** *Left:* Training with a fixed iteration count ($K = 2$) causes a large spike at the first unseen step ($k = 3$). *Right:* Randomized training ($K \in \{2, 3, 4\}$) yields stable, near-exponential decay of $\|H^{(k)} - H^{(k-1)}\|$ and convergence to the numerical floor.

### G.4 EQUIVALENCE: PARALLEL SOLUTION $H^*$ MATCHES SEQUENTIAL RESULT $H_{\text{SEQ}}$

Since the result of the parallel Jacobi iteration $H^{(k)}$ converges to both $H^*$ and $H_{\text{seq}}$, by the uniqueness of limits, we know that $H^* = H_{\text{seq}}$, proving the two are completely identical.

## H RANDOMIZING THE ITERATION COUNT

**Setup.** We compare two schemes for the number of Jacobi iterations used in the hidden-state update: (i) a *fixed iteration* scheme with $K = 2$; and (ii) a *randomized iteration* scheme where $K \sim \text{Unif}\{2, 3, 4\}$. All other training settings are identical. At evaluation, we track the root mean squared change of hidden states $\text{RMSE}(k) = \|H^{(k)} - H^{(k-1)}\|$ over up to 30 iterations.

**Observation.** With *fixed $K = 2$*, the curve exhibits a pronounced spike at the first unseen step ($k = 3$), indicating overfitting to a specific computational depth and poor generalization beyond trained iterations. In contrast, *randomized $K \in \{2, 3, 4\}$* induces a smooth, near-exponential decay that persists well beyond the trained range, converging to the BFloat16 numerical floor.

## I TRAINING DETAILS

The primary computational cost comes from pretraining Ours-1.4B on the 300B-token Pile dataset. This pretraining was conducted on a cluster of high-performance 64GB GPUs and required a total of 73,047 GPU hours.

