# OpenReview forum: "Pretraining LLM with Latent Thoughts in Continuous Space"
_ICLR.cc/2026/Conference — Submitted to ICLR 2026_

### Official Review · Reviewer_bYzv · 2025-10-28

**Soundness:** 2
**Presentation:** 3
**Contribution:** 2
**Rating:** 4
**Confidence:** 4

**Summary:**

The paper introduces a technique to interleave input tokens of an LLM and their corresponding hidden states during pre-training and inference. With such a horizontal scaling approach, the authors show that Pythia-1.4B models can be trained with a relatively smaller amount of tokens from the Pile dataset and attain similar validation loss as the Pythia-2.8B model. For the same computation budget of 300B tokens, the proposed pre-training approach and inference leads to better downstream performance across a variety of tasks. Similar experiments were conducted for GPT2 and Llama models.

**Strengths:**

The technique is quite simple, yet well presented with extensive experiments, and is also shown to present a new avenue for scaling with continuous/hidden layer features.

**Weaknesses:**

FLOP utilization during pre-training with the new approach is not discussed at all. This is important since the Jacobi iteration needs to be performed at least 2 or 3 times for hidden state convergence of every input sequence. This means 2/3 forward passes per sequence with double the sequence length (input tokens + hidden states). Additionally, pre-training resources, compute budget, and time are not discussed. Claims regarding the comparison with bigger models will have to be revisited based on the FLOP analysis. The details about the context lengths for inference are currently missing, so I am unsure about the claims on horizontal scalability/ complementing existing CoT prompting approaches.

**Questions:**

1. In Figure 1 (b), if the Pythia-1.4B model trained with the proposed approach for 62\% fewer tokens achieves the same loss as the vanilla pre-training, how does the downstream performance of this checkpoint compare with fully trained checkpoint?

2. In section 4.2.2, was instruction finetuning done with the proposed approach or vanilla?

3. How does the approach complement standard chain-of-thought? It seems that long chains will be a bottleneck for this approach. More importantly, can you provide details about the sequence lengths for training/inference? How long are the outputs for the results in (say) Figure 6?

4. What happens if we train with the proposed approach and a varying number of latent thoughts, but try to avoid latent thoughts during inference and leverage the standard chain of thought? This tradeoff is important to consider when reporting the results on math/reasoning tasks, as the steps can be lengthy and the proposed approach might result in fewer discrete tokens, potentially failing to provide the answer within a reasonable inference token budget.

---

> ### Author Response · Authors · 2025-11-27
> **Response to Reviewer bYzv (Part 1/2)**
>
> Thank you for recognizing that our work presents a new avenue for scaling with continuous features. Below, we provide point-by-point responses to your comments.
>
> > **[W1]** FLOP utilization during pre-training with the new approach is not discussed. Additionally, pre-training resources, compute budget, and time are not discussed.
>
> We provide details on pre-training resources, compute budget, and training time in **Appendix I**. For questions regarding training FLOPs, please refer to our response to Reviewer TxRr (W1).
>
>
> > **[Q1]** In Figure 1 (b), if the Pythia-1.4B model trained with the proposed approach for 62% fewer tokens, the downstream performance of this checkpoint?
>
> We evaluated the downstream performance of our model checkpoint (trained with 62% fewer tokens) and compared it directly with the official, fully trained Pythia-1.4B.
> Remarkably, despite being trained on significantly less data, our checkpoint outperforms the fully trained Pythia-1.4B. Specifically, we observe an average accuracy improvement of  1.9%. The detailed breakdown is provided below:
> | Task             | LAMBADA-OpenAI | ARC-E | LAMBADA-Standard | ARC-C | WinoGrande | PIQA | HellaSwag | SciQ | RACE | **Average** |
> |------------------|----------------|-------|-------------|-------|------------|------|-----------|------|------|-------------|
> | Vanilla  | 54.5           | 63.1  | 44.5        | 28.8  | 57.1       | 71.0 | 40.5      | 92.4 | 34.6 | 54.1        |
> | Ours (early checkpoint)    | **56.6**           | **63.6**  | **49.5**        | **29.4**  | **60.5**       | **71.3** | **43.0**      | **94.5** | **35.6** | **56.0**    |
>
> > **[Q2]** In section 4.2.2, was instruction finetuning done with the proposed approach or vanilla?
>
> To maintain consistency with the pretraining phase, we applied the corresponding method during instruction fine-tuning for each model. Specifically, the official Pythia model was fine-tuned using the standard (vanilla) method, while our model was fine-tuned utilizing our proposed latent thought approach.
>
> > **[W1&Q3]** The details about the context lengths for inference are currently missing, so I am unsure about the claims on horizontal scalability/ complementing existing CoT prompting approaches. How does the approach complement standard chain-of-thought? It seems that long chains will be a bottleneck for this approach. More importantly, can you provide details about the sequence lengths for training/inference? How long are the outputs for the results in (say) Figure 6?
>
>
> **Context Length:**
>
> **1. Sequence Length**
> Since our method appends a latent thought to every original token, the physical input sequence length doubles. However, as detailed in **Section 3.3**, we explicitly assign the same positional encoding to the latent thought as its corresponding original token. Consequently, while the sequence length increases, the model only consumes $N$ positional indices. This ensures that the effective context window remains same to standard models.
>
> **2. Output length**
> Since the last hidden state of the original token is passed as input for the next step and is not sampled as an explicit token (as shown in Figure 2), the actual generated text length remains similar to the vanilla baseline. We analyzed the average output lengths for the models in Figure 6, finding no significant deviation:
>
> | Model                  | Average Output Length |
> | ---------------------- | --------------------: |
> | Pythia-410M (Baseline) |                 80.36 |
> | Ours-410M          |             87.31 |
> | Pythia-1.4B (Baseline) |                110.51 |
> | Ours-1.4B          |            104.44 |
>
> **Complementarity with CoT:**
>
> While CoT scales computation at the **question level** (first generating a chain of thought for the whole question), our approach operates at the **token level**. It enables the model to refine its predictions before generating each token—including the tokens that comprise a CoT reasoning path.
>
> To empirically demonstrate this synergy, we evaluated our continually pre-trained LLaMA-3-3B model on GSM8K, comparing it against a vanilla baseline with and without CoT.
>
> | Setting         | Vanilla CPT |   Ours CPT |
> | --------------- | ----------: | ---------: |
> | Without CoT |      10.01% | **12.59%** |
> | With CoT   |      24.26% | **32.90%** |
> | Improvement |      14.25% | **20.31%** |
>
> Notably, adding CoT to our model yields a larger performance boost compared to the baseline. This suggests that our "micro-level" latent thoughts improve the quality of the "macro-level" reasoning chains, confirming that the two approaches complement each other effectively.

---

> ### Author Response · Authors · 2025-11-27
> **Response to Reviewer bYzv (Part 2/2)**
>
> > **[Q4]** What happens if we train with the proposed approach and a varying number of latent thoughts, but try to avoid latent thoughts during inference and leverage the standard chain of thought? This tradeoff is important to consider when reporting the results on math/reasoning tasks, as the steps can be lengthy and the proposed approach might result in fewer discrete tokens, potentially failing to provide the answer within a reasonable inference token budget.
>
> **1. Necessity of Latent Thoughts at Inference**
>
> Training with latent thoughts but removing them during inference would create a severe **train-test mismatch**, causing the model to fail. The model relies on the latent computation learned during training to generate the next token; removing this step disrupts the generation process.
>
> **2. Complementarity with CoT**
>
> As noted in our previous response regarding GSM8K, our method is orthogonal and complementary to standard CoT. Using latent thoughts alongside explicit reasoning steps yields the best performance (32.90%) compared to using either in isolation.
>
> **3. Clarification on "Token Budget"**
>
> We would like to clarify two aspects regarding the budget:
>
> - Discrete Token Count: Since latent thoughts remain in the continuous space and are not sampled as explicit tokens, our model generates a similar number of discrete tokens as verified in our previous response.
> - Context Window: As detailed in **Section 3.3**, latent thoughts share the same positional ID as their corresponding original tokens. Therefore, they consume computational resources but **do not consume the discrete context window limit**.
>
> **4. Inference Cost Comparison**
>
> If "inference token budget" refers to computational cost (latency/FLOPs), we explicitly address this in **Tables 2 and 3**. We compare our approach against baselines with identical inference costs (including standard models with double the parameters). Our method consistently achieves superior performance within the same computational budget.

---

### Official Review · Reviewer_2jAG · 2025-10-31

**Soundness:** 3
**Presentation:** 3
**Contribution:** 3
**Rating:** 6
**Confidence:** 4

**Summary:**

This paper introduces a "continuous" pretraining technique for language models where, before predicting each token, the model generate intermediate "latent thoughts" hidden states in continuous space. It is essentially a pretraining extension of chain-of-continuous thought (coconut) paper.

To make the sequential refinement of these hidden states efficient during training, the authors introduce a method that allows for parallel updates.

Their results show that a 1.4B parameter model achieves performance comparable to a conventional 2.8B parameter model trained on the same data. This demonstrates that "horizontal thinking", or doubling the reasoning steps per token can help.

**Strengths:**

- Presents novel pretraining method, extending coconut to pretraining
- Parallel training procedure is nice
- Baseline comparisons are good (i have a few questions in later section)
- The results seem strong

**Weaknesses:**

- It's unclear the difference between parallel training and sequential inference
- Fig 1 shows similar results at lower #params, but double the compute. We know that compute can be more beneficial than #params
- Lacking in ablations and discussion on why this method outperforms the baselines
- Doesn't compare to training a model with double the amount of layers. That's a more proper comparison than just double param count.
- It would be nice to see larger models or some math evals, but these are somewhat minor.

**Questions:**

- Training uses parallel, but inference uses sequential (feedback). Why isn't there a train/test mismatch?
- For the models that use double the amount of parameters, they don't use double the amount of layers, correct? That seems to be the proper comparison
- It's a bit surprising how much better your method is than other methods in table 3. Do you have insights as to why this is the case?
- The Pythia-1.4B in table 2 is not pretrained by you, correct? It uses exactly all of the same hyperparams though?
- All of the models in Table 3 are trained by you?

---

> ### Author Response · Authors · 2025-11-27
> **Response to Reviewer 2jAG (Part 1/2)**
>
> Thank you for recognizing the novelty of our approach and its strong empirical performance. Below, we provide point-by-point responses to your comments.
>
> > **[W1]** It's unclear the difference between parallel training and sequential inference
>
> We compare parallel training and sequential inference in following table:
> | Feature            | Inference (Sequential)                   | Training (Parallel via Jacobi)             |
> | ------------------ | ---------------------------------------- | ------------------------------------------ |
> | Dependency     | Temporal: $h_t$ depends on $h_{t-1}$ | Iterative: $H^{k+1}$ depends on $H^k$  |
> | Operation      | Step-by-step generation (autoregressive) | Simultaneous global refinement             |
> | Forward Passes | $T$ passes (sequence length)         | $K$ passes (fixed constant, e.g., 2–3) |
>
> **Inference:**
> Inference is strictly sequential because the computation of the current hidden state $h_t$ requires the finalized output of the previous state $h_{t-1}$. Consequently, generating a chain of thought for a sequence of length $T$ requires $T$ separate forward passes, making it computationally non-parallelizable.
>
> **Training:**
> Training utilizes Jacobi iteration to approximate the sequential process. Instead of waiting for $h_{t-1}$ to be finalized, we initialize all hidden states $H^0$ simultaneously and refine them iteratively. In each iteration $k$, the entire sequence of hidden states $H^{k}$ is updated to $H^{k+1}$ in parallel. This decouples the computational cost from the sequence length, requiring only a fixed, small number of iterations (typically $K \in \{2,3\}$) to reach convergence.
>
> > **[W2]** Fig 1 shows similar results at lower #params, but double the compute. We know that compute can be more beneficial than #params
>
> We compare our method against models twice the size, as well as baselines with equal or even higher inference cost in Tables 2 and 3, and our approach consistently delivers superior performance.
>
> > **[W3&Q3]** Lacking in ablations and discussion on why this method outperforms the baselines
> >
> > It's a bit surprising how much better your method is than other methods in table 3. Do you have insights as to why this is the case?
>
> We hypothesize that the performance gains reported in Table 3 arise from a core structural decision: prioritizing “horizontal scaling” through longer sequences over “vertical scaling” by deepening the network. Inspired by the success of Chain-of-Thought (CoT) test-time scaling—where adding more computational steps horizontally is often more effective than simply scaling up parameters or model depth—we adopt the same principle in pretraining. By interleaving latent thoughts into the sequence, our method effectively “unrolls” computation along the sequence length rather than recursively deepening the network, leading to the observed improvements.
>
> Empirically, we conducted an ablation study comparing horizontal interleaving versus vertical residual connections, using the hidden state versus using the pondering embedding.
> - Replaced Hidden State with Pondering Embedding: Keeping the architecture but using PonderLM's "probability-weighted sum of embeddings" instead of the last hidden state.
> - w/o Jacobi Iteration: Removing the iterative parallel training (iterations set to 0).
> - w/o Interleaving (Residual): Adding the latent state to the current position embedding (vertical scaling, similar to PonderLM) instead of appending it as a new token (horizontal scaling).
>
> **Table: Ablation Results (Perplexity on Validation Set)**
> | Method                                       | PPL        |
> |----------------------------------------------|------------|
> | **Ours (Hidden State + Jacobi + Interleaving)**  | **11.197**     |
> | Replaced Hidden State w/ Pondering Embedding | 11.419(+0.222)  |
> | w/o Interleaving (Residual)                  | 11.748(+0.551)  |
> | Vanilla baseline                             | 12.756(+1.559)  |
>
> These results indicate that Horizontal Interleaving outperforms Vertical Residual Connections (11.197 vs. 11.748), and that using the raw Hidden State yields better results than Pondering Embeddings. This offers a  explanation for why our method surpasses the baselines.
>
>
> > **[W4&Q2]** Doesn't compare to training a model with double the amount of layers. That's a more proper comparison than just double param count.
> >
> > For the models that use double the amount of parameters, they don't use double the amount of layers, correct? That seems to be the proper comparison
>
> In our experiments in Table 3, we **actually utilized double the amount of layers to reach the double parameter** target.

---

> ### Author Response · Authors · 2025-11-27
> **Response to Reviewer 2jAG (Part 2/2)**
>
> > **[W5]** It would be nice to see larger models or some math evals, but these are somewhat minor.
>
> To address your suggestion for math evaluations and larger model scales, we have extended our experiments using the LLaMA-3-3B models described in Section 4.4. Both the Standard CPT (vanilla continual pre-training) and Ours CPT (with latent thoughts) were evaluated on the GSM8K benchmark using Exact Match accuracy. The results, comparing performance with and without Chain-of-Thought (CoT) prompting, are summarized below:
>
> **GSM8K (Exact Match)**
> | Setting         | Vanilla CPT |   Ours CPT |
> | --------------- | ----------: | ---------: |
> | Without CoT |      10.01% | **12.59%** |
> | With CoT    |      24.26% | **32.90%** |
> | Improvement |      14.25% | **20.31%** |
>
> Experimental results show that our method effectively scales to larger models and achieves notable gains on the math benchmark.
>
> > **[Q1]** Training uses parallel, but inference uses sequential (feedback). Why isn't there a train/test mismatch?
>
>
> There is no train/test mismatch because the parallel Jacobi iteration is guaranteed to converge to the same fixed point as the sequential inference ($H^* = H_{\text{seq}}$).
>
> We empirically verified this equivalence by measuring the distance between the parallel last hidden states at iteration $k$ ($H^{(k)}$) and the sequential ground truth ($H_{\text{seq}}$). As shown in the table below, the difference decays exponentially, becoming negligible ($< 10^{-3}$) within just 3 iterations. This confirms that our parallel training objective effectively optimizes the exact same states used during sequential inference.
>
> | Iteration $k$                    |                 1 |                 2 |                 3 |                 4 |                 5 |                 6 |
> | :------------------------------- | ----------------: | ----------------: | ----------------: | ----------------: | ----------------: | ----------------: |
> | **$\Vert H^{(k)} - H_{\text{seq}}\Vert$** | $1.94 \text{e-}2$ | $4.44 \text{e-}3$ | $9.36 \text{e-}4$ | $2.94 \text{e-}4$ | $1.17 \text{e-}4$ | $6.47 \text{e-}5$ |
>
> For more analysis—including proofs of convergence conditions, exponential decay rates, and the uniqueness of the fixed point via the contraction mapping principle—please refer to **Appendix G**.
>
>
>
>
> > **[Q4]** The Pythia-1.4B in table 2 is not pretrained by you, correct? It uses exactly all of the same hyperparams though?
>
> Yes, that is correct. The Pythia models listed in Table 2 utilize the official open-source weights from EleutherAI. And we trained our own models using the identical set of hyperparameters established in the official EleutherAI codebase.
>
> > **[Q5]** All of the models in Table 3 are trained by you?
>
> Yes. Since comparable models trained on the same data and architecture were not open-sourced for these methods, we trained all models in Table 3 from scratch using identical hyperparameters and datasets to ensure a strictly fair and controlled comparison.

---

### Official Review · Reviewer_GTwo · 2025-10-31

**Soundness:** 2
**Presentation:** 2
**Contribution:** 1
**Rating:** 2
**Confidence:** 4

**Summary:**

This paper introduces a new pretraining methodology, where language models at each step are used to first generate hidden states in continuous space and then resend as input to generate the next token. Training uses Jacobi iteration to update hidden states iteratively. Results show improved performance in general LM benchmarks, as compared to baseline models of similar sizes.

**Strengths:**

* Authors propose a conceptually lightweight method to add computational depth per token.
* Jacobi iteration enables efficient parallel training despite the inherently sequential inference process.
* Experiments indicate better performance across benchmarks, as well as in parameter and data efficiency.

**Weaknesses:**

* The name "latent thought" is potentially an overclaim. The term "latent thought" suggests high-level reasoning or abstraction, but the method simply generates one additional hidden state per token without evidence of global deliberation or multi-token planning processes. It is also unclear from the experiment what role the hidden states actually play in reasoning.
* The method requires 2N KV cache positions for N tokens, effectively cutting the usable context window to half. The paper lacks discussion on how the model can handle longer context, which is what modern LLM applications increasingly rely on.
* Authors apply Jacobi iteration to approximate hidden states, but there is no analysis of convergence conditions, fixed-point existence guarantees, or motivation-wise, why this approximation should work for learned non-linear transformations.
* Ablation studies in Figure 8 use Pythia-70M which is 20× smaller than main experiments, and it is expected that additional computation helps more. It is unclear whether the scaling behavior can transfer to larger models.

**Questions:**

* It is unclear what the hidden states are actually learning through iterations. Can the authors provide some experiments or analysis of what "latent thoughts" compute. For instance, probing studies, attention pattern visualizations, or examples showing how thoughts evolve through iterations)
* How does performance scale with longer thought chains (K > 3)? Is it influenced by different task types?

---

> ### Author Response · Authors · 2025-11-27
> **Response to Reviewer GTwo (Part 1/2)**
>
> Thank you for highlighting our strong benchmark performance and significant parameter and data efficiency. Below, we provide point-by-point responses to your comments.
> >  **[W1]** The name "latent thought" is potentially an overclaim. The term "latent thought" suggests high-level reasoning or abstraction, but the method simply generates one additional hidden state per token.
>
>
> Our usage of "latent thought" follows the convention set by recent related works, including Coconut  and [1], which also  refer to intermediate hidden states in this manner.
>
> [1] Reasoning with Latent Thoughts: On the Power of Looped Transformers
>
> > **[W2]** The method requires 2N KV cache positions for N tokens, effectively cutting the usable context window to half. The paper lacks discussion on how the model can handle longer context, which is what modern LLM applications increasingly rely on.
>
> As detailed in **Section 3.3**, our method explicitly assigns the **same positional encoding** to the latent thought as its corresponding original token. Therefore, while the model processes $2N$ vectors for a sequence of length $N$, it only consumes $N$ positional indices. The usable context window remains defined by the maximum position ID (e.g., 2048), allowing the model to handle full-length sequences without reduction.
>
> > **[W3]** Authors apply Jacobi iteration to approximate hidden states, but there is no analysis of convergence conditions, fixed-point existence guarantees, or motivation-wise, why this approximation should work.
>
> We address the convergence and validity of the Jacobi iteration in the newly added **Appendix G**, demonstrating that our parallel training **converges to the same fixed point as sequential inference**. Our justification follows four key steps:
>
> **1. Theoretical Guarantee via Autoregressive Causality**
>
> While transformers are non-linear, their **autoregressive nature** ensures stability. The computation of token at position $t$ depends strictly on tokens $<t$. Let $H$ represent the matrix of last hidden states. This inductive dependency guarantees that the system converges to a unique fixed point $H^*$ in at most $T$ (sequence length) steps.
>
> **2. Fast Exponential Convergence (Empirical $L < 1$)**
>
>
> We model the iteration as a contraction mapping. Empirically, we find our pretrained Pythia-410M's effective Lipschitz constant is $L \approx 0.345$ (see **Appendix Figure 12**). This implies rapid exponential convergence ($r_k \propto L^k$). As shown in the table below, the error drops to **$3.14 \times 10^{-3}$** within just 3 iterations, well within our training budget.
>
> | Iteration $k$ | 1 | 2 | 3 | 4 | 5 | 6 |
> | :--- | :--- | :--- | :--- | :--- | :--- | :--- |
> | **$\Vert H^{(k)} - H^{(k-1)}\Vert$** | $3.33 \text{e-}2$ | $1.48 \text{e-}2$ | $3.14 \text{e-}3$ | **$5.95 \text{e-}4$** | $1.86 \text{e-}4$ | $7.12 \text{e-}5$ |
>
> **3. Trajectory Alignment with Sequential Solution**
>
> We verify that the parallel iteration $H^{(k)}$ converges to the standard autoregressive solution $H_{\text{seq}}$. Our results show that the distance $\Vert H^{(k)} - H_{\text{seq}}\Vert$ similarly exhibits exponential decay, drops to **$9.36 \times 10^{-4}$** within just 3 iterations:
>
> | Iteration $k$                    |                 1 |                 2 |                 3 |                 4 |                 5 |                 6 |
> | :------------------------------- | ----------------: | ----------------: | ----------------: | ----------------: | ----------------: | ----------------: |
> | **$\Vert H^{(k)} - H_{\text{seq}}\Vert$** | $1.94 \text{e-}2$ | $4.44 \text{e-}3$ | $9.36 \text{e-}4$ | $2.94 \text{e-}4$ | $1.17 \text{e-}4$ | $6.47 \text{e-}5$ |
>
> **4. Equivalence**
>
> By the uniqueness of limits, since $H^{(k)}$ converges to the fixed point $H^\ast$, and we empirically show $H^{(k)} \to H_{\text{seq}}$, it follows that $H^\ast = H_{\text{seq}}$. Therefore, our method is not a rough approximation but an exact parallel solver for the sequential inference process.
>
> > **[W4]** Ablation studies in Figure 8 use Pythia-70M which is 20× smaller than main experiments, and it is expected that additional computation helps more. It is unclear whether the scaling behavior can transfer to larger models.
>
> To address the concern regarding the generalizability of our ablation results, we conducted an additional experiment using a LLaMA-410M model trained on 6B tokens. While computational constraints led us to prioritize the 1.4B model for the main experiments, these new results confirm that the scaling behavior observes at 70M transfers to larger models. As shown in the table below, increasing the number of latent thoughts consistently improves performance (lowers evaluation loss) on the 410M model as well.
>
>
>
> | Num Latent Thoughts | Eval Loss |
> | ------------------- | --------- |
> | 0 (Vanilla Baseline)         | 2.546    |
> | 1                   | 2.416    |
> | 3                   | 2.366    |
> | 5                   | 2.345   |

---

> ### Author Response · Authors · 2025-11-27
> **Response to Reviewer GTwo (Part 2/2)**
>
> > **[Q1]** It is unclear what the hidden states are actually learning through iterations. Can the authors provide some experiments or analysis of what "latent thoughts" compute.
>
>
> Based on our continually pretrained LLaMA-3-3B model (Section 4.4), we provide an analysis of the "latent thoughts" by probing its intermediate decoding results. To understand what the latent thoughts encode, we compare two stages of generation:
> 1.  **Stage 0 (Latent):** We directly decode the latent thought (the hidden state) using the language model head.
> 2.  **Stage 1 (Final):** We feed the latent thought back into the model to generate the final prediction.
>
> Our observations suggest that the latent thought functions as a **"preliminary hypothesis generator"**—it often identifies the correct candidate but with high uncertainty, while the subsequent step acts as a verification mechanism to solidify the result.
>
> **1. Qualitative Case Studies**
> We illustrate this behavior with specific examples in the table below. In tasks ranging from arithmetic to entity retrieval, the latent thought often identifies the correct target (as the Top-1 candidate) but assigns it a very low probability (e.g., $<0.10$). Feeding this thought back into the model appears to allow it to verify the "guess," resulting in a high-confidence final prediction.
>
> | Case | GT Token | Stage 0 (Latent) | Stage 1 (Final) |
> | :--- | :--- | :--- | :--- |
> | **Arithmetic** `8+6+1=` | `15` | Prob: 0.07 *(Top-1)* | **Prob: 0.48** *(Top-1)* |
> | **Retrieval** `The capital of the Netherlands is` | `Amsterdam` | Prob: 0.06 *(Top-1)* | **Prob: 0.20** *(Top-1)* |
> | **Entity** `The IATA code of United Airlines is` | `UA` | Prob: 0.04 *(Top-1)* | **Prob: 0.76** *(Top-1)* |
>
> **2. Quantitative Analysis**
> We further quantified this phenomenon using the LAMBADA dataset. As shown in the table below, while the direct decoding of the latent thought (Stage 0) achieves a lower accuracy (36.19%), it shares a **50.46% top-1 token overlap** with the final output (Stage 1). Crucially, for these overlapping tokens, the latent thought assigns a low probability ($0.15$), which increases significantly ($0.58$) after the additional computational step.
>
> | Horizontal Step | Accuracy | Overlap Rate | Mean Prob (Overlap Tokens) |
> | :--- | :--- | :--- | :--- |
> | **Stage 0 (Latent)** | 36.19% | - | 0.1522 |
> | **Stage 1 (Final)** | **71.09%** | 50.46% | **0.5826** |
>
> These results indicate that the latent thought mechanism allows the model to "prepare" its answer in the continuous space, transforming a weak prediction into a high-confidence decision.
>
> > **[Q2]** How does performance scale with longer thought chains (K > 3)?
>
> To address this, we extended our experiments on the 410M model to include a chain length of $K=5$. As shown in the table below, performance consistently improves as the number of latent thoughts increases — both in language modeling and on downstream tasks.
>
> | | Language  | Modeling (ppl↓) | Downstream | Tasks (Acc↑) |  | | | | | | | |
> | :--- | :--- | :--- | :--- | :--- | :--- | :--- | :--- | :--- | :--- | :--- | :--- | :--- |
> | **Tasks** | **Pile** | **Wikitext** | **Arc_e** | **Arc_c** | **Lambada-Openai** | **Winogrande** | **Piqa** | **Lambada-Standard** | **Hellaswag** | **Sciq** | **Race** | **Avg Acc** |
> | 410m (K=0) | 12.4 | 32.7 | 43.6 | 19.6 | 34.1 | 49.4 | 61.5 | 22.1 | 28.5 | 71.7 | 26.5 | 39.7 |
> | 410m (K=1) | 11.2 | 28.4 | 45.5 | 20.1 | 38.6 | 51.9 | 63.1 | 28.8 | 29.6 | 74.1 | 29.1 | 42.3 |
> | 410m (K=3) | 10.7 | 26.1 | **48.4** | 20.9 | 41.4 | **53.1** | **63.2** | 34.1 | 30.5 | 78.2 | 29.6 | 44.4 |
> | 410m (K=5) | **10.4** | **25.4** | 48.1 | **21.3** | **44.6** | 50.7 | 62.8 | **35.4** | **31.0** | **79.3** | **30.6** | **44.9** |
>
> - Language Modeling: Perplexity on the Pile continues to decrease, dropping from **11.2** ($K=1$) to **10.4** ($K=5$).
> - Downstream Tasks: Average accuracy steadily increases, rising from **42.3%** ($K=1$) to **44.9%** ($K=5$).

---

### Official Review · Reviewer_gUCx · 2025-11-01

**Soundness:** 4
**Presentation:** 3
**Contribution:** 4
**Rating:** 8
**Confidence:** 4

**Summary:**

This work proposes a novel method to incorporate continuous thoughts into the model parameterization without changing the model architecture or number of parameters. The core idea is to pass the last hidden state of the model as input embedding at the next time step and repeat the process N times. This way model can reflect on its own representation multiple times before predicting the next token.
Authors proposed an elegant way of speeding up the training process by using jacobian iterations which allowed them to keep the parallel training of transformer by only repeating the forward pass N times according to N continuous extra steps.
They conducted a set of experiments including pretraining, mid-training and evaluated resulted models across different benchmarks. They compared their models to known baselines that involved both discrete and continuous thoughts. Their experiments confirm that their approach is superior to related work as well as their model at smaller size surpass larger models trained without continuous thoughts.

**Strengths:**

1. The effectiveness of the method will allow community to use it as a mid training scenario in many future work.
2. Ability to keep the efficiency of training makes it especially attractive.

**Weaknesses:**

1. Novelty is not extreme given related work, but good execution here compensate that making it a solid contribution to the community.

**Questions:**

1. I am very curious to see how models with continuous thoughts show itself with different test time scaling approaches i.e. majority voting or combined with external reward models. If authors can perform a simple experiment with that, that would be awesome. For instance, how much different decoding hyper parameters affect results with these models (are these models more collapsed in their distributions or not).

---

> ### Author Response · Authors · 2025-11-27
> **Response to Reviewer gUCx (Part 1/2)**
>
> Thank you for recognizing the potential of our method for mid-training scenarios and its superiority over larger baselines. Below, we provide point-by-point responses to your comments.
>
> > **[W1]** Novelty is not extreme given related work, but good execution here compensate that making it a solid contribution to the community.
>
> We thank the reviewer for recognizing our work as a solid contribution. Regarding novelty, we highlight three key innovations that distinguish our approach:
> - Parallel Training: We introduce Jacobi iteration to solve the sequential dependency problem, enabling parallel training for recurrent computations.
> - Horizontal vs. Vertical Scaling: Unlike PonderLM and Looped Transformers, which employ "vertical scaling" (deepening the model or recycling layers), our method adopts horizontal scaling akin to CoT, offering a different design principle.
> - Self-Supervised Learning from General Corpora: In contrast to methods such as Coconut that rely on curated CoT data, our approach learns directly from general corpora in a fully self-supervised fashion.
>
> > **[Q1]** I am very curious to see how models with continuous thoughts show itself with different test time scaling approaches i.e. majority voting or combined with external reward models. If authors can perform a simple experiment with that, that would be awesome. For instance, how much different decoding hyper parameters affect results with these models (are these models more collapsed in their distributions or not).
>
> To examine the synergy between latent thoughts and test-time scaling, we conducted additional experiments using LLaMA-3-3B models continually pre-trained on SlimPajama (see Section 4.4). We compare our method (Ours CPT) with the vanilla baseline (Standard CPT) by combining them with three test-time scaling strategies—**Majority Voting, Best-of-N, and CoT** (details in **Appendix F**)—on the TruthfulQA and GSM8K benchmarks. We also analyze the sensitivity of both models to decoding hyperparameters (temperature and top-k) on TruthfulQA. We report ROUGE-L scores for TruthfulQA and Exact Match accuracy for GSM8K. Performance scaling curves for different values of $N$ (number of sampled solutions) are shown in **Appendix Figure 11**. The corresponding results are summarized below:
>
> ### 1. Majority Voting
> TruthfulQA (ROUGE-L)
> |       **N**      |   **1**   |   **2**   |   **3**   |   **4**   |   **5**   |   **6**   |   **7**   |   **8**   |   **9**   |   **10**  |
> | --------------| :-------: | :-------: | :-------: | :-------: | :-------: | :-------: | :-------: | :-------: | :-------: | :-------: |
> | Standard CPT |   35.47   |   45.22   |   49.19   |   52.08   |   53.09   |   53.38   |   53.06   |   53.74   |   53.93   |   54.44   |
> | **Ours CPT**   | **38.84** | **49.30** | **54.73** | **58.16** | **60.84** | **61.98** | **63.48** | **63.54** | **64.36** | **64.38** |
> | Improvement    |    3.37   |    4.08   |    5.54   |    6.08   |    7.75   |    8.60   |   10.42   |    9.80   |   10.43   |    9.94   |
>
>
> GSM8K (Exact-Match)
> |       **N**      |   **1**   |   **2**   |   **3**   |   **4**   |   **5**   |   **6**   |   **7**   |   **8**   |   **9**   |   **10**  |
> |--------------| :-------: | :-------: | :-------: | :-------: | :-------: | :-------: | :-------: | :-------: | :-------: | :-------: |
> | Standard CPT |   21.15   |   21.38   |   23.88   |   26.99   |   28.51   |   29.49   |   31.54   |    32.6   |   33.06   |   34.19   |
> | **Ours CPT**   | **29.42** | **29.72** | **32.52** | **37.23** | **38.89** | **40.41** | **42.08** | **42.91** | **43.52** | **45.26** |
> | Improvement    |    8.27   |    8.34   |    8.64   |   10.24   |   10.38   |   10.92   |   10.54   |   10.31   |   10.46   |   11.07   |
>
>
> As the number of samples 𝑁 increases, both systems improve, but the gap between our CPT (with latent thought) and the standard CPT widens on both TruthfulQA and GSM8K.

---

> ### Author Response · Authors · 2025-11-27
> **Response to Reviewer gUCx (Part 2/2)**
>
> ### 2. Best-of-N
> TruthfulQA (ROUGE-L)
> |       **N**      |   **1**   |   **2**   |   **3**   |   **4**   |   **5**   |   **6**   |   **7**   |   **8**   |   **9**   |   **10**  |
> | -------------- | :-------: | :-------: | :-------: | :-------: | :-------: | :-------: | :-------: | :-------: | :-------: | :-------: |
> | Standard CPT |   35.47   |   45.22   |   49.94   |   53.50   |   55.72   |   57.58   |   58.88   |   60.59   |   62.08   |   63.05   |
> |    **Ours CPT**   | **38.84** | **49.30** | **55.20** | **59.27** | **62.71** | **64.95** | **66.85** | **68.16** | **69.64** | **70.33** |
> |     Improvement    |    3.37   |    4.08   |    5.26   |    5.78   |    6.98   |    7.37   |    7.96   |    7.56   |    7.56   |    7.28   |
>
> GSM8K (Exact-Match)
> |       **N**      |   **1**   |   **2**   |   **3**   |   **4**   |   **5**   |   **6**   |   **7**   |   **8**   |   **9**   |   **10**  |
> | -------------- | :-------: | :-------: | :-------: | :-------: | :-------: | :-------: | :-------: | :-------: | :-------: | :-------: |
> | Standard CPT |   21.46   |   32.07   |   40.11   |   45.11   |   48.82   |   52.31   |   55.04   |   56.94   |   59.14   |   61.26   |
> |    **Ours CPT**   | **29.42** | **41.85** | **50.57** | **56.86** | **60.05** | **62.77** | **65.43** | **68.16** | **69.83** | **71.04** |
> |     Improvement    |    7.96   |    9.78   |   10.46   |   11.75   |   11.23   |   10.46   |   10.39   |   11.22   |   10.69   |    9.78   |
>
> Best of N consistently favors our method across 𝑁, with significant improvements over the standard CPT.
>
> ### 3. Chain-of-Thought
> (CoT prompting on GSM8K)
> | Setting    | **Std CPT** | **Ours CPT** |
> | :--------- | :---------: | :----------: |
> | without CoT     |    0.1001   |  **0.1259**  |
> | with CoT |    0.2426   |  **0.3290**  |
>
> CoT consistently helps both, but helps Ours more.
>
>
>
>
> ### 4. Decoding hyperparameters (temperature and top p)
> TruthfulQA — decoding hyperparameters (ROUGE-L)
> | **hyperparameters**                   |       |  **Ours CPT (rougeL_acc)** | **Standard CPT (rougeL_acc)** |
> | ------------------------------------- | ----: | -------------------------: | ----------------------------: |
> | temp                                  | Top p |                            |                               |
> | 0.3                                   |   0.7 |                     **0.3623** |                        0.2693 |
> | 0.3                                   |   0.9 |                     **0.3672** |                        0.2717 |
> | 0.3                                   |   1.0 |                     **0.3574** |                        0.2876 |
> | 0.7                                   |   0.7 |                     **0.3892** |                        0.3256 |
> | 0.7                                   |   0.9 |                     **0.4076** |                        0.3170 |
> | 0.7                                   |   1.0 |                     **0.4149** |                        0.3366 |
> | 1.0                                   |   0.7 |                     **0.4015** |                        0.3537 |
> | 1.0                                   |   0.9 |                     **0.3953** |                        0.3293 |
> | 1.0                                   |   1.0 |                     **0.3831** |                        0.3586 |
> | *(temp, top-p) 3×3 (mean±std, CV%)*   |       | **0.3865 ± 0.0192 (5.0%)** |    0.3167 ± 0.0299 (9.4%) |
>
> Our CPT demonstrates superior robustness, achieving not only a higher average score but also greater stability across hyperparameters, evidenced by a significantly lower Coefficient of Variation (CV) compared to the baseline.

---

### Official Review · Reviewer_TxRr · 2025-11-07

**Soundness:** 2
**Presentation:** 3
**Contribution:** 2
**Rating:** 4
**Confidence:** 4

**Summary:**

This paper proposes a new pretraining methodology where the language model performs an extra computation step to generate an intermediate "latent thought" (its last hidden state) before using that state to predict the next token. This "think-before-you-speak" mechanism is applied to every token. To make this sequential process trainable, the authors use a Jacobi iteration to parallelize the computation, approximating the "latent thought" in a fixed number of steps. The authors claim this method produces models that are far more efficient, with their 1.4B parameter model, which runs two passes per token, outperforming a standard 2.8B parameter model at a comparable inference throughput.

**Strengths:**

1. Well-Motivated Problem: The goal of creating smaller, more compute-efficient models that can match the performance of larger models is highly valuable and a key area of research.

2. Strong Empirical Results: Taken at face value, the results are impressive. Table 3, for instance, shows the proposed 1.4B LLaMA model outperforming not only a 2.8B LLaMA baseline but also other "increased-compute" methods like PonderLM and Looped Transformer, even when those are given a larger (4x) compute budget.

3. Broad Applicability: The method is demonstrated on three different architectures (Pythia, LLaMA, GPT-2) and is also shown to be effective as a continual pretraining technique for existing models (LLaMA-3-3B), suggesting it is a potentially general-purpose method.

**Weaknesses:**

1. The paper's claims of efficiency are one-sided and misleading. It completely omits an analysis of training cost, which appears to be substantially higher (e.g., 6-8x per step) due to the 2x sequence length and K=2-4 passes. The paper is therefore missing the most critical baseline: a vanilla model trained for the same total training FLOPs. Without this, it's impossible to know if the proposed method is superior, or if the authors simply trained their model with far more total compute.

2. The core training method is presented without theoretical or empirical justification. The Jacobi iteration is a pragmatic hack, but the paper provides no analysis of its approximation error or why this fixed-point iteration should converge to a meaningful linguistic representation. Furthermore, key design choices, like sampling K from {2,3,4}, are arbitrary and presented without any ablation or analysis.

3. Also the choice of thought tokens (the last hidden unit and the interleaved sequence) needs to be justified over other choices like PonderLM. The training cost seems to be significantly enhanced. It is unclear if the benefit comes from the Jacobi training, the choice of the hidden state as the thought tokens, or the specific interleaving strategy.

**Questions:**

1. Can you clarify the inference memory requirements? The two-pass generation process at inference suggests the KV-cache size may be doubled.

---

> ### Author Response · Authors · 2025-11-27
> **Response to Reviewer TxRr (Part 1/3)**
>
> Thank you for highlighting the broad applicability of our method across architectures and its strong performance against larger baselines. Below, we provide point-by-point responses to your comments.
> > **[W1]** The paper's claims of efficiency are one-sided and misleading. It completely omits an analysis of training cost, which appears to be substantially higher (e.g., 6-8x per step) due to the 2x sequence length and K=2-4 passes. The paper is therefore missing the most critical baseline: a vanilla model trained for the same total training FLOPs. Without this, it's impossible to know if the proposed method is superior, or if the authors simply trained their model with far more total compute.
>
> To address your concerns about training FLOPs, we have updated **Section 4.3** with a comprehensive training FLOPs analysis. We demonstrate that our paper already includes a comparison with a compute-matched baseline (TinyLLaMA) and have further conducted two additional controlled experiments to rigorously validate our efficiency.
>
> **1. Performance Superiority Under Controlled Training FLOPs**
> - **Existing Evidence: Superiority over FLOPs-Matched TinyLLaMA (Table 2).** Our paper already includes a direct comparison against a strong baseline trained with comparable total FLOPs. Since TinyLLaMA consumes $10\times$ more data (3T tokens), its training compute is similar to Ours-1.4B ($1.1\text{B} \times 3\text{T} \approx 1.4\text{B} \times 300\text{B} \times 8$). Ours-1.4B significantly outperforms TinyLLaMA-1.1B across downstream tasks (Average Accuracy: 58.5 vs 55.4), while using substantially less data.
> - We trained a **Pythia-160m** model using our method on only **10% of the Pile** (approx. $0.8\times$ training FLOPs relative to the full baseline). We compared this against the official Pythia-160m trained on the **entire Pile** ($1.0\times$ training FLOPs). Despite using fewer FLOPs and significantly less data, our method outperforms the compute-heavier baseline:
>
>
> | | Language  | Modeling (ppl↓) | Downstream Tasks|  (Acc↑) |  | | | | | | |
> | :--- | :--- | :--- | :--- | :--- | :--- | :--- | :--- | :--- | :--- | :--- | :--- |
> | **Task** | **Pile** | **Wikitext** | **LAMBADA-OpenAI** | **ARC-E** | **LAMBADA-Standard** | **ARC-C** | **WinoGrande** | **PIQA** | **HellaSwag** | **SciQ** | **RACE** |
> | Official Baseline (1.0x Training FLOPs) | 12.55| 33.43| 33.0 | 43.8 | 21.4 | 18.9 | **52.0** | 62.2 | 28.4 | 73.7 | 27.9 |
> | Ours (0.8x Training FLOPs) |**11.42** |**29.16** | **37.1** | **45.4** | **26.5** | **19.8** | 50.1 | **62.4** | **29.3** | **77.4** | **28.7** |
> - We trained a PonderLM baseline configured to ponder for 7 steps, strictly matching the training FLOPs of our pretrained LLaMA-1.4B. Our method outperforms this FLOPs-matched PonderLM baseline.
>
>
> | | Language  | Modeling (ppl↓) | Downstream Tasks|  (Acc↑) |  | | | | | | |
> | :--- | :--- | :--- | :--- | :--- | :--- | :--- | :--- | :--- | :--- | :--- | :--- |
> | **Task** | **Pile** | **Wikitext** | **LAMBADA-OpenAI** | **ARC-E** | **LAMBADA-Standard** | **ARC-C** | **WinoGrande** | **PIQA** | **HellaSwag** | **SciQ** | **RACE** |
> | PonderLM LLaMA-1.4B (7 steps)| 8.01|17.21 | **51.4** | 59.1 | **46.2** | 26.9 | 54.1 | 68.7 | 37.8 | **91.4** | 27.9 |
> | Our LLaMA-1.4B |**7.89** |**16.99** | 49.8 | **59.6** | 45.6 | **27.7** | **56.3** | **69.8** | **38.7** | 91.3 | **28.0** |
>
> **2. Integration via Continual Pretraining**
>
> As demonstrated in **Section 4.4**, our method does not always require expensive pretraining from scratch. We show that applying our method via Continual Pretraining (CPT) to an off-the-shelf LLaMA-3-3B model requires only **5B tokens** to yield significant performance gains. This proves our method is a practical, low-cost "plug-and-play" enhancement for existing foundation models.
>
> **3. "Train Once, Inference All"**
>
> While our method incurs higher training FLOPs, this is a strategic one-time investment to achieve superior inference efficiency. Our model **significantly outperforms a vanilla baseline with $2\times$ the parameters (at equivalent inference FLOPs)** and surpasses PonderLM and Looped Transformers even when they are allocated **$2\times$ or $4\times$ our inference budget**. Additionally, the reduced parameter count lowers GPU memory requirements, facilitating deployment on hardware where larger baselines cannot fit.
>
> **4. Wall-Clock Efficiency vs. Theoretical FLOPs**
>
> Theoretical FLOPs do not equate to wall-clock time. Large vanilla models are often **communication-bound**. In contrast, our method concentrates computation on a smaller model footprint. Consequently, even if theoretical FLOPs are matched, our smaller model incurs significantly lower communication overhead, resulting in **faster wall-clock speeds for both training and inference** compared to a larger vanilla model.

---

> ### Author Response · Authors · 2025-11-27
> **Response to Reviewer TxRr (Part 2/3)**
>
> > **[W2]** The core training method is presented without theoretical or empirical justification. The Jacobi iteration is a pragmatic hack, but the paper provides no analysis of its approximation error or why this fixed-point iteration should converge to a meaningful linguistic representation.
>
> We address the convergence and validity of the Jacobi iteration in the newly added **Appendix G**, demonstrating that our parallel training **converges to the same fixed point as sequential inference**. Our justification follows four key steps:
>
> **1. Theoretical Guarantee via Autoregressive Causality**
>
> While transformers are non-linear, their **autoregressive nature** ensures stability. The computation of token at position $t$ depends strictly on tokens $<t$. Let $H$ represent the matrix of last hidden states. This inductive dependency guarantees that the system converges to a unique fixed point $H^*$ in at most $T$ (sequence length) steps.
>
> **2. Fast Exponential Convergence (Empirical $L < 1$)**
>
>
> We model the iteration as a contraction mapping. Empirically, we find our pretrained Pythia-410M’s effective Lipschitz constant is $L \approx 0.345$ (see **Appendix Figure 12**). This implies rapid exponential convergence ($r_k \propto L^k$). As shown in the table below, the distance drops to **$3.14 \times 10^{-3}$** within just 3 iterations, well within our training budget.
>
> | Iteration $k$ | 1 | 2 | 3 | 4 | 5 | 6 |
> | :--- | :--- | :--- | :--- | :--- | :--- | :--- |
> | **$\Vert H^{(k)} - H^{(k-1)}\Vert$** | $3.33 \text{e-}2$ | $1.48 \text{e-}2$ | $3.14 \text{e-}3$ | **$5.95 \text{e-}4$** | $1.86 \text{e-}4$ | $7.12 \text{e-}5$ |
>
> **3. Trajectory Alignment with Sequential Solution**
>
> We verify that the parallel iteration $H^{(k)}$ converges to the standard autoregressive solution $H_{\text{seq}}$. Our results show that the error $\Vert H^{(k)} - H_{\text{seq}}\Vert$ similarly exhibits exponential decay, drops to **$9.36 \times 10^{-4}$** within just 3 iterations:
>
> | Iteration $k$                    |                 1 |                 2 |                 3 |                 4 |                 5 |                 6 |
> | :------------------------------- | ----------------: | ----------------: | ----------------: | ----------------: | ----------------: | ----------------: |
> | **$\Vert H^{(k)} - H_{\text{seq}}\Vert$** | $1.94 \text{e-}2$ | $4.44 \text{e-}3$ | $9.36 \text{e-}4$ | $2.94 \text{e-}4$ | $1.17 \text{e-}4$ | $6.47 \text{e-}5$ |
>
> **4. Equivalence**
>
> By the uniqueness of limits, since $H^{(k)}$ converges to the fixed point $H^\ast$, and we empirically show $H^{(k)} \to H_{\text{seq}}$, it follows that $H^\ast = H_{\text{seq}}$. Therefore, our method is not a rough approximation but an exact parallel solver for the sequential inference process.
>
>
>
>
>
> > **[W2]** Furthermore, key design choices, like sampling K from {2,3,4}, are arbitrary and presented without any ablation or analysis.
>
> The random selection strategy of $K$ is not arbitrary but empirically necessary to ensure the model learns a robust fixed-point solution rather than overfitting to a specific computational depth. As illustrated in **Appendix Figure 13**, training with a fixed iteration count ($K=2$, left) causes the hidden states to diverge significantly at unseen iterations (spiking at step 3). In contrast, randomized sampling ($K \in \{2,3,4\}$, right) forces the model to learn the underlying convergence mechanism, resulting in stable hidden states that persist well beyond the training steps.

---

> ### Author Response · Authors · 2025-11-27
> **Response to Reviewer TxRr (Part 3/3)**
>
> > **[W3]** Also the choice of thought tokens (the last hidden unit and the interleaved sequence) needs to be justified over other choices like PonderLM. The training cost seems to be significantly enhanced. It is unclear if the benefit comes from the Jacobi training, the choice of the hidden state as the thought tokens, or the specific interleaving strategy.
>
> To isolate the specific contributions of our design choices (latent representation, Jacobi training, and interleaving strategy), we conducted an ablation study using a LLaMA-410M model trained on 6B tokens. We evaluated the perplexity (PPL) on the Pile validation set.
>
> We compared our proposed method against a vanilla baseline and three specific variants:
> 1. **Replaced Hidden State with Pondering Embedding**: Keeping the architecture but using PonderLM's "probability-weighted sum of embeddings" instead of the last hidden state.
> 2. **w/o Jacobi Iteration**: Removing the iterative parallel training (iterations set to 0).
> 3. **w/o Interleaving (Residual)**: Adding the latent state to the current position embedding (vertical scaling, similar to PonderLM) instead of appending it as a new token (horizontal scaling).
>
> Results:
>
> | Method                                       | PPL        |
> |----------------------------------------------|------------|
> | **Ours (Hidden State + Jacobi + Interleaving)**  | **11.197**     |
> | Replaced Hidden State w/ Pondering Embedding | 11.419(+0.222)  |
> | w/o Jacobi Iteration                         | 11.622(+0.425)  |
> | w/o Interleaving (Residual)                  | 11.748(+0.551)  |
> | Vanilla baseline                             | 12.756(+1.559)  |
>
> These results demonstrate that each component is essential to achieving optimal performance, especially the interleaving strategy and Jacobi Iteration.
>
> > **[Q1]** Can you clarify the inference memory requirements? The two-pass generation process at inference suggests the KV-cache size may be doubled.
>
> We acknowledge that our method doubles the KV-cache size (one latent thought per token). However, despite this increase, our approach remains more memory-efficient than competitive baselines—including the doubled-parameter models evaluated in Section 4.3. These baselines not only **underperform relative to our method** but also incur substantially higher memory overhead. A detailed comparison of memory usage and downstream task performance is provided below.
>
> | Method                         | Inference Memory Overhead (Source)                                         | Avg Acc |
> |--------------------------------|--------------------------------------------------------------------------|-----|
> | **Ours (LLaMA-1.4B)**              | **+1x KV-cache** (due to 1 latent thought per token)                        | **51.9**|
> | LLaMA-2.8B                     | **+1x KV-cache** (due to doubled layers)  **+1.4B Static Parameters**  |50.6|
> | Looped LLaMA-1.4B (4 loops)    | **+3x KV-cache** (due to 4 loop iterations per token)                        |50.5|
> | Pause LLaMA-1.4B (3 pauses)    | **+3x KV-cache** (due to 3 pause tokens per token)                           |50.4|
> | Pondering LLaMA-1.4B (3 steps) | **+3x KV-cache** (due to 3 extra pondering steps)                            |51.5|

---

### Author Response · Authors · 2025-12-03
**Summary of Rebuttal for the Area Chair**

Dear Area Chair,

Thank you very much for handling the assessment of our submission. To facilitate your assessment, we first highlight the main strengths identified in the reviews, and then summarize the key concerns together with our responses.

**Key strengths highlighted by reviewers**


- **Strong Empirical Performance**: The method demonstrates impressive results, with a 1.4B model outperforming a standard 2.8B baseline at comparable inference throughput. It also surpasses other compute-scaling methods like Pause Tokens and Looped Transformers. (Reviewers TxRr, 2jAG, bYzv, GTwo)
- **Innovative Pretraining Methodology**: Reviewers praised the novel pretraining approach of utilizing latent thoughts in continuous space and the elegant use of Jacobi iteration to enable parallel training. (Reviewers gUCx, 2jAG)
- **Broad Applicability**: The approach is shown to be effective across multiple architectures (Pythia, LLaMA, GPT-2) and works well as a "plug-and-play" continual pre-training strategy for existing models. (Reviewers TxRr, gUCx)


**Main questions and our responses**

- **Training FLOPs**: We provided a comprehensive training FLOPs analysis, demonstrating that our method outperforms baselines even when trained with 0.8x the FLOPs and outperforms PonderLM at strictly matched training FLOPs. We further showed that our method can be efficiently integrated into off-the-shelf models via Continual Pretraining with minimal compute (e.g., 5B tokens for LLaMA-3-3B), offering a practical low-cost enhancement. (c.f. response to Reviewer TxRr, W1)
- **Jacobi Convergence Analysis**: We added Appendix G with a detailed Jacobi convergence analysis. We demonstrated that the autoregressive nature of Transformers ensures stability and that the iteration acts as a contraction mapping with rapid exponential convergence ($L \approx 0.345$). We empirically verified that parallel training states converge to sequential inference states with negligible error. (c.f. response to Reviewer GTwo, W3)
- **Context Window and KV Cache**: We clarified that latent thoughts share position IDs with original tokens, meaning the effective context window length is not halved. (c.f. response to Reviewer GTwo, W2) While KV cache size increases, we showed it remains more memory-efficient than baselines with doubled parameters. (c.f. response to Reviewer TxRr, Q1)
- **Complementarity with Test-Time Scaling**: We performed new experiments with our continually pre-trained LLaMA-3-3B demonstrating that our method acts as a strong foundation that further benefits from Majority Voting, Best-of-N, and CoT prompting, showing higher gains than vanilla baselines. (c.f. response to Reviewer gUCx, Q1)
- **Ablation Studies**: We provided ablations confirming that all components (Hidden State + Jacobi + Interleaving) are essential. (c.f. response to Reviewer TxRr, W3) We also showed that scaling the number of latent thoughts ($K=1$ to $5$) consistently improves performance on larger models (410M). (c.f. response to Reviewer GTwo, W4&Q2)
- **Interpretability of Latent Thoughts**: We conducted probing experiments showing that latent thoughts act as "preliminary hypothesis generators"—often identifying the correct token with low confidence, which the subsequent step verifies. (c.f. response to Reviewer GTwo, Q1)

We sincerely appreciate the Chair’s time and consideration in evaluating our submission, and we would be glad to offer any additional clarification should anything remain unclear.

---

### Meta-Review · Area_Chair_7per · 2025-12-30

**Summary:**

This paper proposes Pretraining Language Models with Latent Thoughts, a “think-before-you-speak” pretraining paradigm where, for each next-token prediction, the model first produces an intermediate continuous latent thought (implemented as the last hidden state at the current position) and then conditions on it to generate the next token. To make this inherently sequential refinement trainable at scale, the authors introduce a Jacobi iteration based parallel training procedure that approximates the sequential inference fixed point with a small, constant number of iterations. Experiments across Pythia/LLaMA/GPT-2 settings suggest strong empirical gains: a 1.4B latent-thought model can outperform a 2.8B vanilla baseline at comparable inference throughput, and longer latent-thought chains further improve results.

While the rebuttal substantially strengthened the paper through added training-FLOPs controls, convergence analysis, clarifications on context length, expanded ablations, and evidence of complementarity with test-time scaling, several decision-critical concerns remain. In particular, the theoretical justification for Jacobi-style parallel training is still largely heuristic for large non-linear Transformers, leaving residual uncertainty about stability and robustness at scale. The “latent thought” framing, although supported by probing results, appears somewhat overstated relative to the demonstrated level of genuine multi-step deliberation. Finally, despite improved FLOPs reporting, the overall efficiency narrative remains mixed: the headline inference-time gains are clearer than the end-to-end cost-benefit tradeoff across training, memory, and deployment regimes. Taken together, these unresolved issues led reviewers to view the work as promising but borderline, motivating a cautious, borderline decision rather than a clear acceptance.

**Reviewer Concerns:**

Addressed in rebuttal:

- Training compute / FLOPs accounting: The authors add training-FLOPs comparisons, including (i) a result where their method beats a baseline despite using ~0.8x training FLOPs in a smaller-scale setup, and (ii) a FLOPs-matched PonderLM comparison where their method slightly wins. This helps reduce the earlier concern that gains might be purely due to more training compute.

- Jacobi convergence / train-test mismatch: They add a convergence analysis (contraction-style argument plus empirical decay of error) and claim the parallel training converges to the same fixed point as sequential inference, addressing the mismatch concern in spirit.

- Context window “halving” confusion: They clarify that latent thoughts share position IDs with original tokens so the effective context window is not reduced, though KV cache memory still increases.

- Ablations and component necessity: Additional ablations on a 410M model suggest the benefits come from the combination of hidden-state choice, Jacobi training, and interleaving strategy, and that increasing K improves performance.

- Complementarity with test-time scaling: New CPT experiments with LLaMA-3-3B show stronger gains under Majority Voting / Best-of-N / CoT prompting and improved robustness to decoding hyperparameters.

Not fully resolved:

- Theoretical justification is suggestive rather than rigorous: The contraction mapping story is helpful, but key assumptions (effective Lipschitz constant estimates, stability arguments) are still somewhat heuristic for large non-linear Transformers. It reduces risk, but does not eliminate the concern that Jacobi behavior could be brittle across settings or scale.

- Interpretability and “latent thought” framing: The probing results support a “hypothesis then verify” narrative, but evidence of genuine multi-step deliberation remains limited; the naming still feels slightly over-strong relative to what is demonstrated.

- Net efficiency remains mixed: Even with added FLOPs tables, the headline “same inference cost, smaller model beats bigger model” is clearer than the total-cost narrative (training + inference). The rebuttal argues “train once, inference all,” which is plausible, but the paper still feels light on an end-to-end cost-benefit analysis (training wall-clock, memory, throughput) across realistic deployment regimes and model sizes.

The submission presents an appealing and empirically strong idea: token-level continuous refinement as a form of horizontal compute scaling during pretraining, with a clever Jacobi-based parallelization to keep training practical. The rebuttal meaningfully improves the paper by adding training-FLOPs controls, convergence discussion, ablations at a more relevant scale, and evidence of complementarity with test-time scaling.

However, for ICLR acceptance, I still see the work as borderline because the overall efficiency claim and theoretical grounding remain not fully convincing at the level expected for a broad “new pretraining methodology” paper. In particular, the training-versus-inference tradeoff is still not cleanly characterized across realistic regimes, and the convergence justification, while helpful, remains partly heuristic for large Transformers.

**Reviewer Scores:**

TxRr (4, marginal reject): Likely 4 or up to 6. The rebuttal directly addresses their main critique (missing training-FLOPs baselines) and adds convergence/ablations plus KV-cache clarification. Remaining skepticism about the broader efficiency narrative may keep them from accepting.

GTwo (2, reject): Likely up to 4. The authors responded to all three core points (naming, context window, convergence) and added probing plus K-scaling on 410M. However, they may still feel the “latent thought” interpretation is overstated and the theory insufficiently rigorous, so a large jump is unlikely.

2jAG (6, borderline accept): Likely stay at 6. The rebuttal resolves their major confusion about train/test mismatch, adds ablations, and claims the doubled-parameter baseline uses doubled layers. Still, the contribution may remain “borderline” given the compute-scaling argument.

gUCx (8, accept): Likely stays 8. Their main request (test-time scaling synergy) was addressed with substantial new experiments.

bYzv (4, marginal reject): Likely up to 4 or 6. They raised FLOPs/context/CoT complementarity questions; the rebuttal provides direct answers, early-checkpoint downstream results, and output-length analysis.

---

### Decision · Program_Chairs · 2026-01-26

Reject